# Single-cell analysis of transcription kinetics across the cell cycle

**Samuel O Skinner[1,2,3], Heng Xu[1,2,4], Sonal Nagarkar-Jaiswal[5], Pablo R Freire[6], Thomas P Zwaka[5,7], Ido Golding[1,2,3,4]\***

[1]Verna and Marrs McLean Department of Biochemistry and Molecular Biology, Baylor College of Medicine, Houston, United States; [2]Center for Theoretical Biological Physics, Rice University, Houston, United States; [3]Department of Physics, University of Illinois at Urbana-Champaign, Urbana, United States; [4]Center for the Physics of Living Cells, University of Illinois at Urbana-Champaign, Urbana, United States; [5]Center for Cell and Gene Therapy, Baylor College of Medicine, Houston, United States; [6]Department of Molecular and Cellular Biology, Baylor College of Medicine, Houston, United States; [7]Department for Developmental and Regenerative Biology, Icahn School of Medicine at Mount Sinai, New York, United States

**Abstract** Transcription is a highly stochastic process. To infer transcription kinetics for a gene-of-interest, researchers commonly compare the distribution of mRNA copy-number to the prediction of a theoretical model. However, the reliability of this procedure is limited because the measured mRNA numbers represent integration over the mRNA lifetime, contribution from multiple gene copies, and mixing of cells from different cell-cycle phases. We address these limitations by simultaneously quantifying nascent and mature mRNA in individual cells, and incorporating cell-cycle effects in the analysis of mRNA statistics. We demonstrate our approach on *Oct4* and *Nanog* in mouse embryonic stem cells. Both genes follow similar two-state kinetics. However, *Nanog* exhibits slower ON/OFF switching, resulting in increased cell-to-cell variability in mRNA levels. Early in the cell cycle, the two copies of each gene exhibit independent activity. After gene replication, the probability of each gene copy to be active diminishes, resulting in dosage compensation.

**\*For correspondence:** golding@bcm.edu

**Competing interests:** The authors declare that no competing interests exist.

## Introduction

Gene expression is a stochastic process, consisting of a cascade of single-molecule events (*Coulon et al., 2014*; *Sanchez and Golding, 2013*), which get amplified to the cellular level. A dramatic consequence of stochastic gene expression is that individual cells within a seemingly homogenous population often exhibit significant differences in the expression level of a given gene (*Raj and van Oudenaarden, 2008*). In fact, cell-to-cell variability in expression levels is the most commonly used proxy for the presence and magnitude of stochastic effects (*Elowitz et al., 2002*; *Raj et al., 2006*; *Raser and O'Shea, 2005*). The mapping between stochastic kinetics and population heterogeneity can be made rigorous by making specific assumptions about the kinetics of gene activity and using stochastic theoretical modeling to predict the copy-number statistics of mRNA or protein that would result from these kinetics (*Friedman et al., 2006*; *Raj et al., 2006*; *Shahrezaei and Swain, 2008*; *Thattai and van Oudenaarden, 2001*). The theoretical prediction is then compared to measured single-cell data, to validate the assumptions and estimate kinetic parameters. Using this approach, cell-cell variability in mRNA numbers has been successfully used to demonstrate the

**eLife digest** Scientific investigation requires researchers to use experimental observations to understand the biological process that resulted in these observations. One example is a cellular process called transcription, where the DNA of a gene is copied many times to make molecules of messenger RNA (mRNA), which are later used as instructions to make proteins. Scientists indirectly measure the dynamics of transcription, that is, how often the gene produces mRNA, by counting how many mRNA molecules there are in many individual cells. These numbers are then compared to the predictions made by a mathematical model of transcription, and if the model and experiment agree well, this is interpreted to mean that the model properly describes how often this gene is transcribed.

Unfortunately, this procedure is not straightforward because many factors complicate the relationship between the dynamics of transcription and the number of mRNAs that will be detected in each cell at any one point in time. For example, it is not possible to tell whether a detected mRNA has just been transcribed, or whether it was made hours ago. The age of the cell and how many copies of the template DNA are present also affect the dynamics of transcription. As a result, mRNA measurements may be misinterpreted, leading to wrong conclusions about how highly particular genes are transcribed.

To address this problem, Skinner et al. first improved the experimental measurements by discriminating between mature mRNA and the new mRNA that is still being transcribed. The experiments also measured how much DNA each cell contains, which indicates how old the cell is. These measurements were incorporated into a new mathematical model that is able to predict the dynamics of transcription of particular genes.

Skinner et al. applied their method to two mouse genes called *Oct4* and *Nanog*, which regulate the transformation of embryonic stem cells into other types of cells. The experiments show that both genes can switch between an "on" state where they are being actively transcribed and an "off" state where little or no mRNA is being produced. However, *Nanog* switches between these two states less often than *Oct4*, which results in larger variations between the numbers of mRNAs between different cells. The experiments also show that over the life of the cell, the level of transcription from each copy of the DNA decreases.

Skinner et al.'s approach can be used to refine our knowledge of the transcription of other genes. However, to further improve our understanding of transcription, measurements of other factors will need to be incorporated into the mathematical models.

bursty, non-Poissonian nature of mRNA production in organisms from bacteria to mammals (*Bahar Halpern et al., 2015b*; *Raj et al., 2006*; *Senecal et al., 2014*; *So et al., 2011*; *Zenklusen et al., 2008*), and to decipher how genetic and cellular parameters modulate these kinetics (*Jones et al., 2014*; *Sanchez and Golding, 2013*).

However, the ability to map back mRNA copy-number statistics to transcription kinetics is limited by a number of factors. First, the measured number of mRNA molecules in the cell represents temporal integration over the lifetime of mRNA molecules (*Raj et al., 2006*). And while in bacteria this lifetime is very short (~mins [*Chen et al., 2015*]), in higher organisms it can be as long as hours (*Schwanhäusser et al., 2011*). Consequently, the measured mRNA level is a poor proxy for the instantaneous activity of the gene. Second, the cellular mRNA combines contributions from all copies of the gene of interest—for example, four copies in a diploid cell at G2. Each of these gene copies acts individually and stochastically (*Hansen and van Oudenaarden, 2013*; *Levesque et al., 2013*); their combined contribution depends on whether they are correlated and how. Finally, the sampled population typically contains a mixture of cells at different phases of the cell cycle. As a result, deterministic changes in gene copy number and activity along the cell cycle add to the measured population heterogeneity, and may be erroneously interpreted as resulting from stochastic effects (*Zopf et al., 2013*).

Here we demonstrate how these limitations can be overcome, such that mRNA statistics is reliably used to infer the kinetic parameters of stochastic gene activity. Specifically, we investigate the

transcriptional activity of *Oct4* and *Nanog*, two key genes in the pluripotency network of mouse embryonic stem cells (*Young, 2011*). Elucidating the stochastic kinetics of these genes, and how it changes along the cell cycle, is crucial for understanding pluripotency and the path to differentiation. For one, *Nanog* expression has been reported to exhibit large cell-to-cell variability (*Filipczyk et al., 2013*; *Kalmar et al., 2009*; *Singer et al., 2014*), and this variability was argued to play an important role in differentiation (*Abranches et al., 2014*; *Chambers et al., 2007*; *Silva et al., 2009*), but both the sources and consequences of *Nanog* variability are still unclear (*Cahan and Daley, 2013*; *Torres-Padilla and Chambers, 2014*). It has also been shown that human stem cells' propensity to differentiate varies significantly between different phases of the cell cycle (*Gonzales et al., 2015*; *Pauklin and Vallier, 2013*; *Singh et al., 2013*), but again, we are lacking a detailed picture of the underlying transcriptional activity of key pluripotency factors along the cell cycle.

To elucidate *Oct4* and *Nanog* kinetics along the cell cycle, we simultaneously measured the numbers of nascent (actively transcribed) and mature mRNA for each gene in individual cells, and used the DNA contents of the cell to determine its cell-cycle phase. We next used the single-cell data to test how gene activity depends on the presence of other copies of the same gene and how it changes as the gene replicates during the cell cycle. This information allowed us to construct a stochastic model for gene activity, which explicitly accounts for the presence of multiple gene copies and the progression of the cell cycle. We then used the cell-cycle-sorted single-cell data to calibrate the theoretical model and estimate the kinetic parameters that characterize *Oct4* and *Nanog* activity.

## Results and discussion

Our first goal was to measure simultaneously nascent and mature mRNA from the genes of interest. While both mRNA species reflect the same underlying kinetics of gene activity, the two are subject to very different kinetics of elimination: Nascent mRNA is eliminated (by being converted to mature mRNA) once elongation and splicing are complete, typically in a few minutes (*Coulon et al., 2014*; *Martin et al., 2013*). In contrast, mature mRNA is subject to active degradation, with a typical half-life of a few hours (*Sharova et al., 2009*). A consequence of these very different time scales is that simultaneously measuring both species for the same gene would allow us to better constrain the theoretical model of gene activity and estimate the underlying parameters (see below and *Figure 1—figure supplement 1*).

To detect nascent and mature mRNA in individual cells, we used single-molecule fluorescence in situ hybridization (smFISH) (*Femino et al., 1998*; *Raj et al., 2008*; *Skinner et al., 2013*) to label the gene of interest, with spectrally-distinct probes sets for the intron and exon sequences (*Hansen and van Oudenaarden, 2013*; *Senecal et al., 2014*; *Vargas et al., 2011*). Under this labeling scheme, nascent mRNA are expected to be bound by both probe sets, while mature mRNA will only exhibit exon-probe binding (*Figure 1A*). Consistent with these expectations, *Oct4* and *Nanog* labeling in mouse embryonic stem cells revealed numerous diffraction-limited spots containing exon-only signal (*Figure 1B*, *Figure 1—figure supplement 2*). In the same cells, only a small number of nuclear spots contained both intron and exon signals (*Figure 1B*, *Figure 1—figure supplement 2*). Neither type of spot was observed in Fibroblasts, where *Oct4* and *Nanog* are not expressed (*Chambers et al., 2003*; *Pesce et al., 1998*) (*Figure 1B*, *Figure 1—figure supplement 2*). We used automated image analysis to identify individual mRNA spots, allocate them to cells and discard false positive spots (*Skinner et al., 2013*) (*Figure 1C*, *Figure 1—figure supplement 3*, Materials and methods 5). We identified the fluorescence intensity corresponding to a single mature mRNA (*Skinner et al., 2013*; *Zenklusen et al., 2008*) and used this intensity value to convert the total fluorescence of exon spots in each cell to the numbers of nascent and mature mRNA (*Figure 1G*). Our measured values for both the mean and coefficient of variation for *Nanog* mRNA per cell ($126 \pm 24$ and $0.80 \pm 0.05$, respectively; designates mean $\pm$ SEM throughout; 3 experiments with >600 cells per experiment; *Figure 1D*) are in excellent agreement with the literature (*Abranches et al., 2014*; *Faddah et al., 2013*; *Grün et al., 2014*; *Hansen and van Oudenaarden, 2013*; *Muñoz Descalzo et al., 2013*; *Ochiai et al., 2014*; *Singer et al., 2014*) (*Supplementary file 1A*). For *Oct4*, our measured mean ($477 \pm 67$; 3 experiments with >700 cells per experiment; *Figure 1D*) is higher than in previous reports (*Faddah et al., 2013*; *Grün et al., 2014*; *Singer et al., 2014*) while our coefficient of variation ($0.34 \pm 0.01$) is in agreement with previous estimates (*Faddah et al., 2013*; *Grün et al., 2014*;

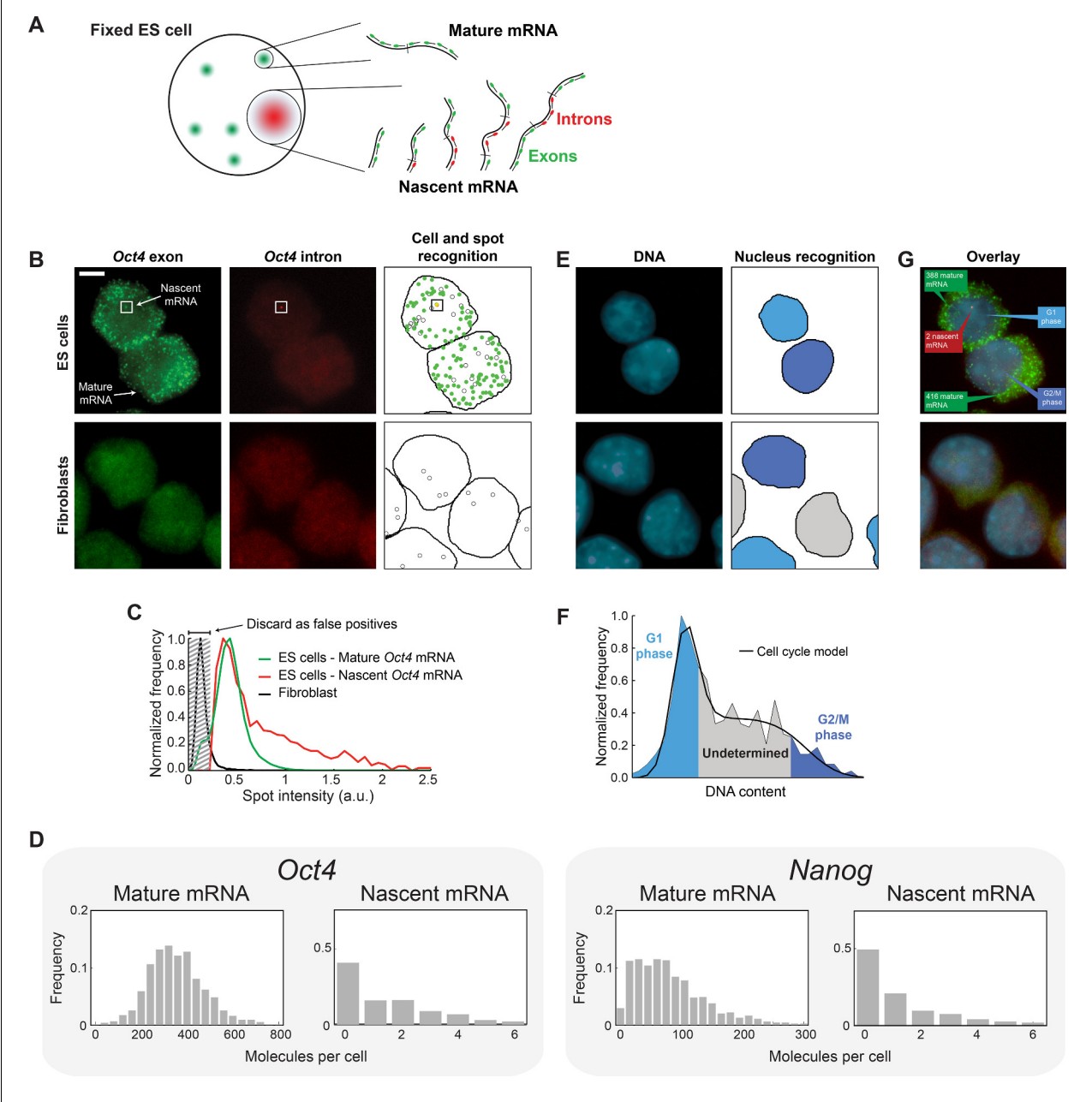

**Figure 1.** Quantifying mature mRNA, nascent mRNA and cell-cycle phase in individual mouse embryonic stem (ES) cells. (**A**) Identifying nascent and mature mRNA. Introns (red) and exons (green) were labeled using different colors of smFISH probes. In the cell, pre-spliced nascent mRNA at the site of active transcription are bound by both probe sets, whereas mature mRNA are only bound by the exon probe set. (**B**) Mouse embryonic stem (ES) cells (top row) labeled for *Oct4* exons (left column, green) and introns (center column, red). Automated image analysis (right column) was used to identify the cell boundaries (black line), intron (red) and exon (green) spots, as well as false-positive spots (black circles, see Panel C). Co-localized exon and intron spots (yellow) were identified as nascent mRNA (square), whereas spots found only in the exon channel were identified as mature mRNA. Fibroblasts (bottom row) were used as negative control. Scale bar, 5 µm. (**C**) The distribution of *Oct4* mRNA spot intensities for mature mRNA (green, >100000 spots), nascent mRNA (red, >1000 spots), and spots found in Fibroblasts (black, >1000 spots). The histograms were used to discard false positive spots (gray region) and to identify the signal intensity corresponding to a single mRNA. (**D**) The distributions of mature and nascent mRNA numbers per cell for *Oct4* (>700 cells) and Nanog (>1000 cells). (**E**) The same cells as in panel B, labeled for DNA using DAPI (left column, blue). Automated image analysis (right column) was used to identify the nuclear boundary (black line). The DNA content of each nucleus was used to estimate the phase of the cell cycle (cyan, grey, and blue shading; see Panel F). (**F**) The distribution of DNA content per cell (>700 cells), estimated from the nuclear DAPI signal (panel E). The histogram of DNA content per cell was fitted to a theoretical model of the cell cycle (black line), and used to identify which cells are in G1 phase (cyan) and which in G2 (blue). (**G**) Overlay of the smFISH and DAPI channels for mouse embryonic stem cells (top) and

*Figure 1 continued on next page*

*Figure 1 continued*

fibroblasts (bottom). The estimated number of mature (green) and nascent (red) mRNA, as well as the phase of the cell cycle (blue), are indicated for the two stem cells.

The following figure supplements are available for figure 1:

**Figure supplement 1.** Fitting both nascent and mature mRNA constrains model parameters.

**Figure supplement 2.** smFISH images of *Nanog* mRNA in ES cells and Fibroblasts.

**Figure supplement 3.** Distribution of *Nanog* mRNA spot intensities.

**Figure supplement 4.** 3D reconstruction of nuclei from the DAPI channel.

**Figure supplement 5.** Fitting the DNA-content histogram to a cell-cycle model.

---

*Singer et al., 2014*) (*Supplementary file 1A*). The difference in mean values may reflect differences in cell lines or experimental conditions.

Next, to identify the cell-cycle phase of individual cells, we used the total DNA contents of each cell, estimated from the DAPI signal integrated over the three-dimensional nucleus (*Figure 1E*, *Figure 1—figure supplement 4*). The distribution of DNA contents from the cell population was well described by the Fried/Baisch model for the cell cycle (*Johnston et al., 1978*) (*Figure 1—figure supplement 5*). We therefore used the model to classify the cells into G1, S and G2/M phases (*Figure 1F,G*). Below we refine this analysis further by calculating, for each cell, its temporal position within the cell cycle and the gene copy number of *Oct4* and *Nanog* (see Figure 3). At this stage, however, we could already identify sub-populations of cells at the G1 and G2 phases of the cell cycle (*Figure 1F*), and use these cells to address the questions of gene-copy independence and dosage compensation.

First, we tested whether individual copies of the same gene act independently of each other, rather than in a correlated manner. To do so, we examined cells in G1, where each gene exists in two copies per cell. We measured the number of nascent mRNA at each copy of the gene. For both *Oct4* and *Nanog*, we did not detect significant correlation between the nascent mRNA levels of the two gene copies in the cell ($r$, Pearson correlation coefficient; *Oct4*: $r = 0.05 \pm 0.04$, p>0.05; *Nanog*: $r = 0.07 \pm 0.01$, p>0.05; 3 experiments with >200 cells per experiment) (*Figure 2—figure supplement 1*). Furthermore, we found that, for both genes, the numbers of active transcription sites per cell followed a binomial distribution, consistent with the assumption that the two copies of the gene act independently of each other (*Figure 2A*; $\chi^2$ goodness of fit test (*Singer et al., 2014*) gives p>0.05 for both *Oct4* and *Nanog*; 3 experiments with >200 cells per experiment). Thus, our data indicate independent stochastic activity of each copy of the gene.

We next wanted to test how the activity of *Oct4* and *Nanog* changes when each of the genes replicates during the cell cycle. Under the simplest assumption, each gene copy will maintain its transcriptional activity irrespective of the total number of gene copies in the cell. In that case, the prediction would be that the total amount of nascent mRNA doubles between G1 and G2 phases (Note that the mature mRNA, due to its much longer lifetime (*Supplementary file 1*), is not expected to immediately follow the gene dosage in such a simple manner; *Figure 3—figure supplement 1*). However, when we compared the nascent mRNA level between G1 and G2 phases, we found that, for both *Oct4* and *Nanog*, the fold change was significantly lower than two (*Oct4*: $1.28 \pm 0.09$, *Nanog*: $1.51 \pm 0.15$; 3 experiments with >200 cells per phase in each experiment; *Figure 2B*). Thus, *Oct4* and *Nanog* exhibit dosage compensation in their activity, analogous to the effect seen for X-chromosome genes between male and female (*Heard et al., 1997*), as well as for some autosomal genes when their copy number is altered (*FitzPatrick et al., 2002*; *Gupta et al., 2006*). The change in gene activity between G1 and G2 was manifested in a <2 fold increase in the number of active transcription sites per cell, while the number of nascent mRNA per active site remained unchanged (*Figure 2B*). In contrast to *Oct4* and *Nanog*, a reporter gene expressed from a strong synthetic promoter (*Niwa et al., 1991*; *Vintersten et al., 2004*) did not show dosage compensation,

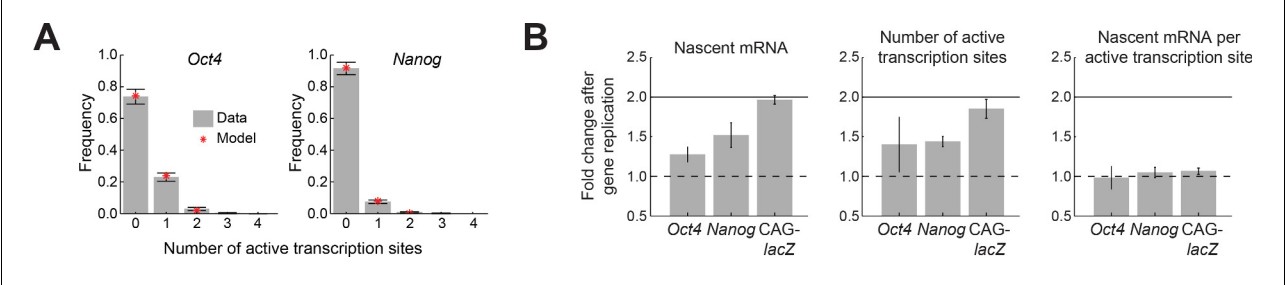

**Figure 2.** *Oct4* and *Nanog* exhibit independent allele activity and dosage compensation. (**A**) The distribution of number of active transcription sites for *Oct4* (left; >700 cells) and Nanog (right; >1,000 cells), in cells having two copies of each gene. In both cases, the measured distribution (gray) is described well by a theoretical model assuming independent activity of the two alleles (binomial distribution, red). Error bars represent the estimated SEM due to finite sampling. (**B**) The fold change in transcriptional activity following gene replication for *Oct4, Nanog*, and a control reporter gene (CAG-*lacZ*). For *Oct4* and *Nanog*, the average number of nascent mRNA (left) increases less than two-fold following gene replication, while a two-fold increase is observed in the control reporter gene. The change in number of nascent mRNA reflects an increase in the number of active transcription sites (middle), with no change in the number of nascent mRNA at each transcription site (right). Error bars represent SEM from 3 experiments with >200 cells per cell-cycle phase in each experiment.

The following figure supplement is available for figure 2:

**Figure supplement 1.** Nascent mRNA correlation between two gene copies.

instead exhibiting a two-fold increase in nascent mRNA following gene replication (1.97 ± 0.07; 2 experiments with >200 cells per phase in each experiment; *Figure 2B*).

To extract the kinetics of *Oct4* and *Nanog* from our single-cell data, we constructed a theoretical model describing the stochastic activity of each gene (*Figure 3A*). In the model, each copy of the gene switches stochastically between active ('ON') and inactive ('OFF') states, with rates $k_{ON}$ and $k_{OFF}$. In the active state, transcription is initiated, again stochastically, with rate $k_{INI}$. Following initiation, nascent mRNA remains at the transcription site for a finite residence time $\tau_{RES}$, representing the combined duration of transcript elongation, splicing and release (*Coulon et al., 2014*; *Hoyle and Ish-Horowicz, 2013*). The nascent mRNA is then deterministically converted into mature mRNA. Mature mRNA is degraded stochastically with rate $k_D$. The copy number of each gene doubles from two to four at a time $\tau_{REP}$ during the cell cycle. Following gene replication, the rate of gene activation $k_{ON}$ changes by a factor $\alpha$, to allow for dosage compensation. Finally, at the end of the cell cycle, mature mRNA are partitioned binomially between the two daughter cells (*Golding et al., 2005*; *Rosenfeld et al., 2005*).

To compare our single-cell data with model predictions, we first mapped the DNA content of each cell to the cell's temporal position within the cell cycle (*Figure 3—figure supplement 2*). This was done using ergodic rate analysis (*Kafri et al., 2013*), which uses static single-cell measurements from steady-state populations to obtain temporal information. We then plotted, for both *Oct4* and *Nanog*, the amount of nascent mRNA as a function of time along the cell cycle (*Figure 3B*). Fitting the data to a step function allowed us to estimate the gene replication time, $\tau_{REP}$, and the fold change in gene activity, $\alpha$. For both genes, the two parameters were consistent with the estimates using G1 and G2 phases, obtained earlier (*Figure 3—figure supplement 3* and *Figure 3—figure supplement 4*).

Next, we proceeded to estimate the kinetic parameters of gene activity for *Oct4* and *Nanog*. For a given set of parameters, we solved the model above using a modified version of the finite state projection algorithm (*Munsky and Khammash, 2006*), extended to include the deterministic process of mRNA elongation, the contribution of multiple gene copies, and the progression of the cell cycle (Materials and methods 8). Solving the model yielded the copy-number distribution for both nascent and mature mRNA at different times along the cell cycle (*Figure 3C*). We then used maximum-likelihood estimation (*Neuert et al., 2013*) to obtain the values of $k_{ON}$, $k_{OFF}$, $k_{INI}$ and $\tau_{RES}$ (*Supplementary file 2A*). For both *Oct4* and *Nanog*, the estimated parameters provided a good fit between model predictions and the experimental histograms (*Figure 3C*). The parameter values

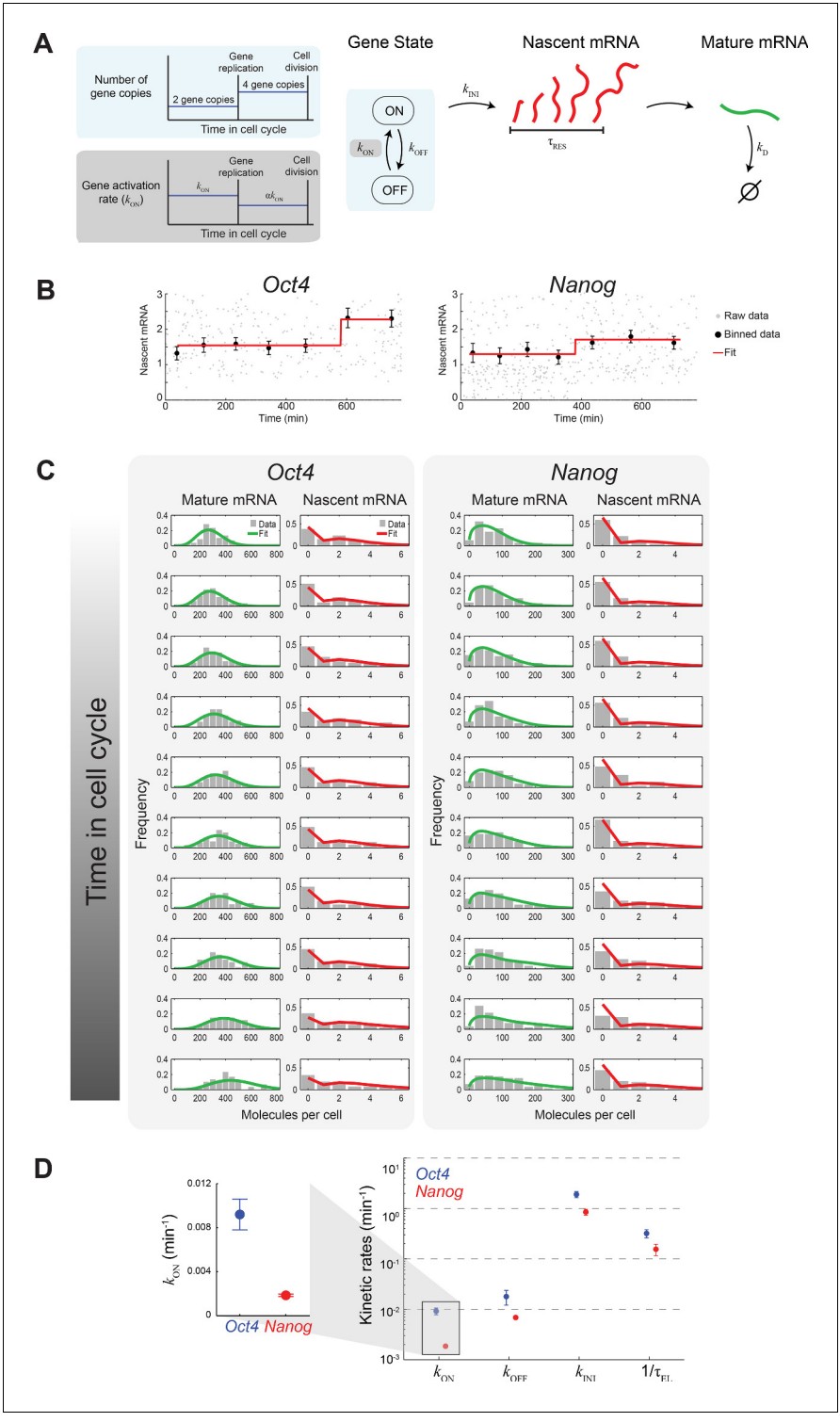

**Figure 3.** Extracting the stochastic kinetics of *Oct4* and *Nanog*. (**A**) A stochastic 2-state model for gene activity, which incorporates cell cycle and gene copy-number effects. Each gene copy stochastically switches between 'ON' and 'OFF' states. Transcription is stochastically initiated only in the 'ON' state. After initiation, the nascent transcript (red) elongates with constant speed, and is then converted into a mature mRNA molecule (green). Mature mRNA are degraded stochastically. Gene copies are independent, and their number changes from 2 to 4 following gene replication (left, cyan box). At the end of the cell cycle, mRNA molecules are binomially partitioned between the two daughter cells. Dosage compensation is included though a decrease in the rate of activation following gene replication (left, grey box). (**B**) Estimating the gene replication time and the fold-change in

*Figure 3 continued on next page*

*Figure 3 continued*

transcriptional activity for *Oct4* (left; >700 cells) and *Nanog* (right; >1000 cells). The number of nascent mRNA was plotted against the time within the cell cycle for each cell (grey points), and the data were binned into populations of equal cell number (black markers). The binned data were fit to a step function (red), used to estimate the gene replication time and the fold-change in number of nascent mRNA before/after gene replication. Error bars represent SEM. (**C**) The distribution of mature and nascent mRNA copy number over time, for *Oct4* (left; >700 cells) and *Nanog* (right; >1000 cells). The cell population was partitioned into 12 time windows, equally-spaced within the cell cycle (rows; we discarded the first and last windows, where the low cell numbers lead to a large error in the ERA calculation [*Kafri et al., 2013*]). The measured distributions (gray) are overlaid with the model predictions for mature (green) and nascent (red) mRNA. (**D**) The probabilistic rates of the transcription process and the gene elongation rate, for *Oct4* (blue) and *Nanog* (red). The rates were estimated from the best theoretical fit of the mature and nascent mRNA distributions (panel C). The rate that varies most between *Oct4* and *Nanog* is the probability of switching to an active transcription state, $k_{ON}$, which is ~5-fold higher for *Oct4* (inset). Error bars represent SEM from 3 experiments with >600 cells per experiment.

The following figure supplements are available for figure 3:

**Figure supplement 1.** Expected behavior of mature and nascent mRNA numbers over time.

**Figure supplement 2.** Mapping DNA content to time in the cell cycle using ergodic rate analysis.

**Figure supplement 3.** Agreement between methods of measuring dosage compensation.

**Figure supplement 4.** Estimated gene replication times fall within S phase.

**Figure supplement 5.** The effect of model representation of dosage compensation on the estimated rates of transcription.

---

were also consistent with previous estimates, in cases where such estimates existed (*Supplementary file 1B*).

What are the kinetics revealed by the model? The *Oct4* and *Nanog* genes spend a comparable fraction of time in the active transcriptional state (*Oct4*: $k_{ON}/(k_{ON}+k_{OFF}) \approx 34\%$ for each gene copy prior to gene replication; *Nanog*: 22% *Supplementary file 2B*). During each of these 'ON' periods, *Oct4* and *Nanog* produce, on average, similar numbers of mRNA (*Oct4*: $k_{INI}/k_{OFF} \approx 110$, Nanog: 120). However, where the two genes vary significantly is in the probabilistic rates of switching between the 'ON' and 'OFF' states, with *Nanog* switching more slowly in both directions ($k_{ON} \approx 9\times10^{-3}$ min$^{-1}$ for *Oct4*, $2\times10^{-3}$ min$^{-1}$ for *Nanog*; $k_{OFF} \approx 2\times10^{-2}$ min$^{-1}$ for *Oct4*, $7\times10^{-3}$ min$^{-1}$ for *Nanog*). In particular, the ~5-fold difference in $k_{ON}$ results in a correspondingly longer average "OFF" duration for *Nanog* (in G1, $\tau_{OFF} = 1/k_{ON} \approx 8.9$ hr, compared to 1.8 hr for *Oct4*; *Supplementary file 2B*).

The differences in transcription kinetics between *Oct4* and *Nanog* also lead, unavoidably, to different degrees of cell-to-cell variability in mRNA numbers. In particular, the higher measured coefficient of variation for *Nanog* (0.80, compared to 0.34 for *Oct4*) is a direct reflection of the lower value of $k_{ON}$ (*Raj et al., 2006*). In other words, the large heterogeneity in *Nanog* levels, highlighted in previous studies (*Abranches et al., 2013*; *Chambers et al., 2007*; *Filipczyk et al., 2013*; *Kalmar et al., 2009*) does not require invoking more complex kinetics than those of other genes (e.g. additional kinetic steps [*Neuert et al., 2013*; *Senecal et al., 2014*]), but merely a difference in the value of a single parameter.

Following gene replication, both *Oct4* and *Nanog* exhibit a decrease in the transcriptional activity of each gene copy. The effect of this dosage compensation is to equalize gene expression along the cell cycle and decrease the degree of cell-to-cell variability. The lower variability may be physiologically significant, as it has been reported that changes in *Oct4* levels as small as two-fold may lead to different cell fates (*Niwa et al., 2000*). The compensatory effect is achieved through a decrease in the probability of each gene copy to be active (0.72 fold for *Oct4* and 0.76 fold for *Nanog*; *Supplementary file 2A*). Similar behavior was recently reported for a number of genes in cultured mammalian cells (*Padovan-Merhar et al., 2015*). These authors also found that the cell volume

(independently of the cell cycle phase) strongly affects the number of nascent mRNA at each transcription site. In our study, the cell-to-cell variability in volume within each cell-cycle phase was significantly smaller than that seen by (*Padovan-Merhar et al., 2015*) (CV≈0.2 versus ≈0.5), preventing us from exploring the effect of cell volume on gene activity. Interestingly, the synthetic reporter gene CAG-*lacZ* did not exhibit dosage compensation. Perhaps the viral enhancer elements included in the promoter (*Niwa et al., 1991*; *Vintersten et al., 2004*) are more resistant to the regulatory mechanisms that create the compensatory effect in endogenous genes.

We note that despite the complex stochastic kinetics of transcription, and the multiple ways that these kinetics can be modulated (*Sanchez et al., 2013*; *So et al., 2011*), some simple unifying features emerge. When comparing the activity of *Oct4* and *Nanog*, we found that the kinetic parameter that varies the most between the two is the probabilistic rate of switching to the active state, $k_{ON}$, while the rates of gene inactivation and of transcription initiation are much closer (*Figure 3D*). The dosage compensation effect following gene replication, observed in both *Oct4* and *Nanog* (*Figure 2B*), is also consistent with a change in $k_{ON}$. These two observations extend a number of recent studies in a range of systems (including one of *Nanog* in mouse embryonic stem cells [*Ochiai et al., 2014*]), all suggesting that varying expression level—along the cell cycle (*Padovan-Merhar et al., 2015*), between different growth conditions (*Ochiai et al., 2014*), or under regulation by a transcription factor (*Senecal et al., 2014*; *Xu et al., 2015*)—is achieved by changing $k_{ON}$. The mechanistic basis for this prevalent phenomenology is yet to be elucidated (*Padovan-Merhar et al., 2015*; *Sanchez and Golding, 2013*).

We have shown how changes in gene copy number and in promoter activity along the cell cycle can be incorporated into the analysis of mRNA copy-number statistics. However, multiple additional factors may contribute to mRNA heterogeneity. First, as noted above, the cell volume has recently been shown to dramatically affect transcription kinetics (*Padovan-Merhar et al., 2015*). Consequently, cell-cell variability in volume will translate into different mRNA levels. Second, the stochastic kinetics of mRNA processing downstream of transcription—splicing (*Coulon et al., 2014*), export from the nucleus (*Bahar Halpern et al., 2015a*; *Battich et al., 2015*), degradation and partition at cell division (*Huh and Paulsson, 2011*)—will too add to mRNA number heterogeneity. Finally, cell-to-cell differences in relevant kinetic parameters—of transcription and the subsequent mRNA processes, of the cell cycle, etc. (so called 'extrinsic noise')—will also contribute to the observed mRNA heterogeneity. Additional work, both experimental and theoretical, is required to delineate the relative contribution of all these factors to the eventual mRNA statistics that we measure.

## Materials and methods

### 1. Cell lines and culture conditions

#### 1.1 Cell lines

Wildtype R1 mouse embryonic stem (ES) cells (ATCC No. SCRC-103) were obtained from Andras Nagy (Mount Sinai Hospital, Lunenfeld, Canada). Z/Red mouse ES cells (*Vintersten et al., 2004*) express *βgeo* (*lacZ* and neomycin-resistance fusion) under the control of a CAG promoter (chicken β-actin promoter coupled with the cytomegalovirus immediate early enhancer) (*Niwa et al., 1991*). Z/Red cells were obtained from Richard R. Behringer (MD Anderson Cancer Center, Houston, TX, USA). NIH-3T3 mouse embryonic fibroblasts were obtained from ATCC (ATCC no. CRL-1658) and used as negative controls.

#### 1.2 Media and growth conditions

ES cells were cultured in Dulbecco's Modified Eagle's High Glucose GlutaMAX Pyruvate Medium (Invitrogen, Carlsbad, CA, 10569) supplemented with 10% fetal bovine serum (FBS; Gemini, West Sacramento, CA, 900-108H), 2 mM L-Glutamine (Gibco, Carlsbad, CA, 25030–081), 100 nM nonessential amino acids (Invitrogen, 11140–050), 0.1 mM β-mercaptoethanol (Fluka, St. Louis, MO, 63690), and 1000 U/ml LIF (Millipore, Billerica, MA, ESG1107). ES cells were grown on 10-cm culture dishes (Corning, Corning, NY, 430167) coated with 0.1% gelatin (Sigma, St. Louis, MO, G1890). NIH-3T3 cells were cultured in Dulbecco's Modified Eagle's high glucose Medium (Gibco, 11965) supplemented with 10% fetal bovine serum, and 1 mM sodium pyruvate (Gibco, 11360). NIH-3T3 cells were grown on 15-cm culture dishes (Corning, 430599).

## 2. Single-molecule fluorescence in situ hybridization

Our protocol is based on Raj et al. (*Raj et al., 2008*). Modifications were made to adapt the protocol to a suspension of mouse embryonic stem cells. Sterile, nuclease-free, aerosol-barrier pipette tips were used. Diethylpyrocarbonate (DEPC)-treated water (Ambion, Carlsbad, CA, AM9922) was used whenever the protocol calls for water.

### 2.1 Probe design and labeling

Nucleic acid sequences with annotations of exons and introns were obtained from the National Center of Biotechnology Information (NCBI) gene database for *Oct4* (GeneID: 18999) and *Nanog* (GeneID: 71950). All exon regions were used as the target sequences for the exon probe set design. Intron 1 of *Oct4* and intron 2 of *Nanog* were used as the target sequences for the intron probe set design. Target intron and exon sequences were searched for species-specific repeats and aligned to the *Mus musculus* RefSeq RNA database using the 'more dissimilar sequences' program in Basic Local Alignment Search Tool (BLAST, NCBI). Any species-specific repeats or similar sequences (alignment score $\geq$80) were removed from the target sequences.

DNA oligonucleotide probes were designed, ordered, and stored following (*Skinner et al., 2013*). In brief, the online program developed by Arjun Raj (*Raj et al., 2008*) (singlemoleculefish.com) was used to design a set of oligonucleotide probes (*Supplementary file 3*) that are complementary to the target sequences. Each probe was ordered with a 3' amine group (mdC(TEG-Amino); Biosearch, Novato, CA). Upon arrival, the oligo solutions were thawed, transferred to separate 1.5-ml microcentrifuge tubes, and stored at -20°C.

The amine-modified oligonucleotide probes were conjugated to succinimidyl-ester-modified dyes following (*Skinner et al., 2013*). *Oct4* exon, *Nanog* exon, and *lacZ* probes sets were labeled with 6-Carboxytetramethylrhodamine (6-TAMRA; Invitrogen, C6123). *Oct4* and *Nanog* intron probe sets were labeled with Alexa Fluor 647 (Invitrogen, A-20006). After labeling, the working stocks of the probe sets were 10–16 µM and had an estimated labeling efficiency of >90% (*Skinner et al., 2013*). The stocks were wrapped in aluminum foil and stored at -20°C.

### 2.2 Sample fixation and permeabilization

ES cells were grown in a 10-cm culture dish coated with 0.1% gelatin to ~80% confluency. The growth medium was aspirated away from the culture dish. The cells were washed twice with 5 ml PBS (Invitrogen, 14190–250) by gently pipetting PBS onto the dish and aspirating. 3 ml of pre-warmed 0.05% trypsin (Invitrogen, 25300–054) was added to the dish to cover the cells. The culture dish was incubated at 37°C for 5 min to allow for trypsin protease activity to create a single-cell suspension. 7 ml of growth medium was added to the culture dish to deactivate the trypsin. The 10 ml of cell suspension was gently pipetted up and down 10 times and transferred to a 15-ml centrifuge tube. The cells were centrifuged at 1200 rpm for 5 min, and the supernatant was aspirated. Cell fixation was performed by resuspending the cells in 5 ml PBS (RNase free; Ambion, AM9625) + 3.7% (v/v) formaldehyde (Ambion, BP531-500) followed by room temperature incubation for 10 min. The cells were centrifuged at 500 g for 5 min and the supernatant was removed. The cells were then resuspended in 5 ml RNase-free PBS, centrifuged at 500 g for 5 min, and the supernatant was removed. The cells were permeabilized by resuspension in 5 ml 70% (v/v) ethanol and incubated at 4°C for 12–16 hr. Finally, the cell density was calculated by washing 25 µl of cells in 300 µl RNase-free PBS and determining the cell count with a hemocytometer. The number of cells obtained from a 10-cm plate was typically ~4x$10^7$ cells, equivalent to a cell density of ~8x$10^6$ cells/ml after permeabilization.

### 2.3 Hybridization and washing

All centrifugation steps were performed at 500 g for 5 min at 4°C. After permeabilization, a volume containing ~1x$10^6$ cells was transferred to a new 1.5-ml microcentrifuge tube. 500 µl of PBST (RNase-free PBS + 0.1% (v/v) Tween 20 [Fisher Scientific, Waltham, MA, BP337-100]) were added to the cells. The cells were pelleted by centrifugation, and the supernatant was removed. The cells were resuspended in 500 µl PBST, pelleted by centrifugation, and the supernatant was removed. The cells were resuspended in 500 µl of wash solution (see below) and incubated at room temperature for 5 min. The cells were then centrifuged and the supernatant was removed. 2 µl of a probe

stock solution was added to 50 µl of hybridization solution (see below). The cells were then resuspended in this hybridization mix and left at 30°C overnight.

In the morning, 500 µl of wash solution was added to the tube and mixed well by pipetting. The tube was incubated at 30°C for 30 min. The cells were pelleted by centrifugation and the supernatant was removed. The cells were washed three more times (i.e. resuspended in 500 µl of wash solution, incubated at 30°C for 1 hr, pelleted by centrifugation, and supernatant removed). 4′,6-diamidino-2-phenylindole (DAPI, Fisher Scientific, PI-46190) was added to the wash solution in the last wash, to a final concentration of 10 µg/ml. The cells were resuspended in 50 µl of 2× SSC (Ambion, AM9763) and kept at 4°C until imaging (less than 24 hr).

## 2.4 Hybridization and washing solutions

Following (*Raj et al., 2006*), a range of formamide concentrations was initially tested to empirically determine the optimal value. 20% (w/v) formamide gave the best results in that it was high enough so that background noise due to non-specific binding was low, while still low enough so that the fluorescence signal from target mRNA molecules was not impaired.

10 ml of wash solution contains 1.76 ml of formamide (Ambion, AM9342), 1 ml of 20× SSC (Ambion, AM9763), and 10 µl Tween-20 (Fisher Scientific, BP337-100). Wash solution was made fresh and stored on ice until use. 10 ml of hybridization solution contains 1 g of dextran sulfate (Sigma, D8906), 1.76 ml of formamide, 10 mg of *E. coli* tRNA (Sigma, R4251), 1 ml of 20× SSC, 40 µl of 50 mg/ml BSA (Ambion, AM2616), and 100 µl of 200 mM ribonucleoside vanadyl complex (New England Biolabs, Ipswich, NY, S1402S). Hybridization solution was filter sterilized and aliquots of 500 µl were stored at -20°C.

## 3. Fluorescence microscopy

### 3.1 Slide preparation

1× PBS + 1.5% agarose pads were prepared following (*Skinner et al., 2013*), and stored between two microscope slides at 4°C for up to 24 hr. For use in imaging, the slides were carefully moved, exposing 1 cm of the agarose pad. A 1 × 1-cm agar pad was excised with a razor blade and placed on a 22 × 22-mm #1 coverslip (Fisher Scientific, 12-545B). 10 µl of cell suspension were pipetted onto the 1 × 1-cm agar pad and incubated in the dark at room temperature for 5 min to allow excess liquid to absorb into the agarose pad. The 22 × 22-mm #1 coverslip with agarose and sample was then inverted and placed at the center of a 24 × 50-mm #1 coverslip (Fisher Scientific, 12-545F).

### 3.2 Microscopy equipment

The samples were imaged using an inverted epifluorescence microscope (Nikon, Melville, NY, Eclipse Ti) equipped with a cooled EM-CCD camera (Photometrics, Tucson, AZ, Cascade II:1024) and motorized stage control (Prior, Rockland, MA, Proscan III). A mercury lamp was used as the light source (Nikon, Intensilight C-HGFIE) with band-pass filter sets (Cy3, Nikon Instruments, 96323; Cy5, Nikon Instruments, 96324; DAPI, Nikon Instruments, 96310). A fast motorized optical shutter (Sutter Instruments, Novato, CA, SmartShutter) was used to control the fluorescence illumination exposure time. A 40×, 1.30 numerical aperture, oil-immersion differential interference contrast (DIC) objective (Nikon, MRH01400) was used with an additional 2.5× lens in front of the camera. The coverslip containing the sample was mounted on a universal specimen holder. The microscope was installed on an optical table (TMC, Peabody, MA, breadboard and four-post support) to dampen mechanical vibrations. 'Elements' software (Nikon) was used to control the microscopy setup. The same imaging protocol was also used with an alternative camera (Photometrics, Evolve 512).

### 3.3 Imaging configuration

The exposure time and gain were chosen such that the maximum pixel value for the fluorescent foci was no higher than 60% of the maximum pixel value of the camera (65535 for a 16-bit camera). Exposure times above 300 ms were avoided to minimize photobleaching. Image stacks consisting of nineteen focal positions with 500 nm spacing were acquired for DIC, Cy5, Cy3, and DAPI images. Each sample was imaged at multiple slide positions to obtain a total of at least 600 cells.

## 4. Nucleus and cell segmentation

We developed custom software in MATLAB to perform nucleus and cell segmentation. For each cell in the fluorescence image stacks, we reconstructed the nucleus in the DAPI channel and recognized the cell boundary in the Cy5 (intron) channel (*Figure 1*, *Figure 1—figure supplement 4*).

To begin reconstructing individual nuclei, a series of morphological operations was performed on each focal plane in the DAPI channel image stack. First, a Sobel filter was applied to obtain the edges of the nuclear slices (the portions of the nuclei visible within the focal plane). Second, morphological filling was applied to fill the interiors of the nuclear slices. Third, the focal plane was smoothed using morphological opening. Finally, the optimal threshold value was determined for each nuclear slice following (*Xu et al., 2015*). Briefly, a series of increasing threshold values was applied. At each threshold value, the area ($A$) and circularity ($4\pi A/P^2$; where $P$ is the perimeter length) of the thresholded nuclear slice was calculated. Once the area and circularity satisfied the criteria: $A > 500$ pixels and $4\pi A/P^2 > 0.7$, the threshold value was used. The processed individual focal planes were stacked to form a 3-dimensional mask. Individual nuclei were identified in the mask as overlapping nuclear slices from neighboring planes.

To recognize the cell boundary, we thresholded the Cy5 (intron) channel because it primarily had two levels of pixel values corresponding to 1) non-specific labeling and/or autofluorescence within cells and 2) the non-cell background. The threshold value was determined using Otsu's method. The reconstructed nuclei were used to segment joined cells using a watershed algorithm with the nuclei as basins, and to remove above-threshold objects that did not contain nuclei. For each image stack, the output of the nucleus and cell segmentation program was visually inspected and refined using a graphical user interface.

## 5. mRNA quantification

### 5.1 smFISH spot recognition and quantification

We used the MATLAB-based spot-recognition software, Spätzcells (*Skinner et al., 2013*) (available for download: https://code.google.com/p/spatzcells/), to identify smFISH fluorescence foci (spots) in image stacks (*Figure 1B*). In brief, local maxima were accepted as potential spots if the pixel value difference between the local maximum and its neighbors was greater than a threshold value. This threshold value was determined empirically by visually inspecting the spot-recognition results from a subset of images. The spots were then matched between focal planes, allowing for a two-pixel shift in $xy$ location. For each spot, the focal plane in which it had the highest intensity was determined. Using the lsqcurvefit function in MATLAB, this focal plane was used to fit the spot, and its potential neighboring spots, to a function consisting of multiple 2-dimensional Gaussians and a tilted plane, of the form:

$$f(x,y) = \sum_{i=1}^{n} A_i e^{-\left[a_i(x-x_i)^2 + b_i(y-y_i)^2 + 2c_i(x-x_i)(y-y_i)\right]} + B_0 + B_x(x-x_0) + B_y(y-y_0)$$

where $n$ is the number of spots in the neighborhood of the central spot, $A_i$ is the amplitude of each Gaussian, $a_i, b_i, c_i$ are the elliptical shape parameters of each Gaussian, $x_i, y_i$ are the $xy$ locations of each Gaussian, $B_0, B_x, B_y$, define the height and orientation of the tilted plane, and $x_0, y_0$ define the center of the fitting area (*Skinner et al., 2013*). The integrated intensity of a single spot was calculated as the integral over the single Gaussian function: $I_i = \frac{A_i \pi}{\sqrt{a_i b_i - c_i^2}}$.

Following (*Skinner et al., 2013*), we discarded false positives by comparing the spot intensity (Gaussian amplitude, $A$; *Figure 1C*, *Figure 1—figure supplement 3*) of the spots in the negative control sample to the ones in the positive sample. A 'false-positive threshold' was selected in spot intensity that separated the population of false positives from the population of genuine spots in the positive sample. Spots with intensity lower than the 'false-positive threshold' were discarded from the subsequent analysis of all samples (*Figure 1C*, *Figure 1—figure supplement 3*).

To identify the value of integrated intensity that corresponds to a single mRNA molecule, a histogram of integrated intensities ($I$) was constructed using the spots above the 'false-positive threshold' in the exon channel (*Figure 1C*, *Figure 1—figure supplement 3*). Following the strategy of (*Skinner et al., 2013*; *Zenklusen et al., 2008*), this histogram was then fitted to a sum of Gaussians, where each Gaussian in the sum has a mean equal to integer multiples of the first, representing

multiple mRNA in each spot. The mean of the first Gaussian was estimated as the typical integrated intensity of a single mRNA molecule. For each spot, this value was then used to convert the integrated intensity to the number of mRNA molecules (*Skinner et al., 2013*; *Zenklusen et al., 2008*).

Each spot was assigned to a cell using the cell masks obtained earlier (Materials and methods 4). We calculated the total number of mRNA in a cell by summing over the number of mRNA in all spots assigned to that cell. The total number of mRNA consists of the numbers of mature mRNA and nascent mRNA at active transcription sites.

## 5.2 Identification of active transcription sites and quantification of nascent mRNA

We identified active transcription sites in the exon channel as spots that matched intron-channel spots, allowing a two pixel shift in $xy$ dimensions and 1 focal plane shift (*Figure 1B*, *Figure 1—figure supplement 2*; criteria used previously by [*Hansen and van Oudenaarden, 2013*]). Exons spots that were not matched were assumed to be mature mRNA. The number of nascent mRNA at each active transcription site was quantified in the exon-channel by dividing the integrated intensity by the integrated intensity of a single-mRNA molecule (Materials and methods 5.1). Each active transcription site was assigned to a cell using the cell masks (Materials and methods 4). We calculated the number of nascent mRNA in a cell by summing over the nascent mRNA at all active transcription sites in that cell. When testing for the independence of allele activity (*Figure 2A*), we followed (*Hansen and van Oudenaarden, 2013*) and only counted transcription sites with >1 nascent mRNA. We then fitted the distribution of number of active transcription sites to a binomial distribution (*Figure 2A*).

## 6. DNA quantification and cell-cycle phase determination

### 6.1 Quantification of DNA content

To quantify the DNA content in individual cells, we used the nuclear and cell masks created previously (Materials and Methods 4; *Figure 1*, *Figure 1—figure supplement 4*), which define the boundary of each nucleus and cell. For each cell, the total DAPI fluorescence ($D$) was calculated as the sum of the DAPI-channel pixel values within the nuclear boundary, and the volume ($V$) was calculated as the total number of pixels in the nucleus. The background of the DAPI image ($b$) was calculated as the median DAPI pixel value of the non-cell pixels in the cell mask. For each cell, the DNA content was calculated as: $DNA = D - bV$.

### 6.2 Fitting the DNA-content distribution to a cell-cycle model and determining cell-cycle phases

The distribution of DNA contents was fitted using the Fried/Baisch model (*Johnston et al., 1978*) (*Figure 1F*, *Figure 1—figure supplement 5*), which approximates the DNA content distribution as a superposition of Gaussians with equal coefficients of variation (CV = μ/σ, the ratio of the mean to the standard deviation). In this model, the DNA content of the cells in G1 phase is approximated as a Gaussian distribution with mean $\mu$ and standard deviation $\sigma$. The DNA of cells in G2/M phases is approximated as a Gaussian distribution with mean $2\mu$ and standard deviation $2\sigma$. The DNA of cells in S phase is approximated as the sum of three Gaussian distributions each with CV's equal to that of the G1 Gaussian. The cell cycle model has the form:

$$f(x) = \sum_{i=1}^{5} A_i e^{-\left(\frac{x - \alpha_i \mu}{\sqrt{2}\alpha_i \sigma}\right)^2}, \ \alpha_i = (i+3)/4$$

where $f(x)$ is the frequency of observing a cell with DNA content, $x$. The fitting parameters of this function are: the G1 Gaussian mean $\mu$, the G1 Gaussian width $\sigma$, and heights of the Gaussian distributions associated with each stage: $A_1$ for G1 phase, $A_2$, $A_3$, and $A_4$ for S phase, and $A_5$ for G2/M phases. This model was able to accurately describe the measured distribution of DNA content (*Figure 1F*, *Figure 1—figure supplement 5*).

To investigate features of cells in G1 or G2/M phases, where cells have two and four copies of autosomal genes, respectively, we determined ranges of DNA content that correspond to cells in G1 phase or in G2/M phases. To determine the desired ranges of DNA content for each experiment,

we used the fit of the cell-cycle model to the DNA content histogram and the extracted fit parameters, $\mu$ and $\sigma$ (**Figure 1F**). We observed that the cell-cycle model describes large ranges of DNA contents that contain a mixed population of cell cycle phases (**Figure 1—figure supplement 5**), so we sought to determine DNA content values that would minimize the overlap of the phases predicted by the model. By visually inspecting the model fit results, we determined that the following gating satisfied those aims: Cells with DNA content less than $\mu + \sigma$ were categorized as cells in G1 phase, while cells with DNA content more than $2\mu$ were categorized as cells within G2/M phases (**Figure 1F**, **Figure 1—figure supplement 5**). Using this analysis, we estimated the fraction of cells in G1, S, and G2/M phases to be 43 ± 2%, 29 ± 4%, and 28 ± 4%, respectively (mean ± SEM from 6 experiments with >600 cells per experiment).

## 7. Using ergodic rate analysis to extract temporal information

### 7.1 Using ergodic rate analysis to calculate the time within the cell cycle

The ergodic rate analysis (ERA) transform described in (**Kafri et al., 2013**) was developed to extract temporal dynamics from measurements of a fixed steady-state population. In the current work, we used the ERA transform to map the measured DNA content $x$ to the time $t$ within the cell cycle for each cell (**Figure 3—figure supplement 2**). To do so, we transformed the DNA content distribution (fitted using the cell-cycle model, Materials and methods 6.2 above), $f(x)$, as follows:

$$t(x) = \tau_{DIV} \log_2 \left( \frac{2}{2 - F(x)} \right)$$

where $F(x) = \int_0^x f(x')dx'$ is the cumulative DNA distribution. The timescale in this calculation is introduced using the doubling time of the cells, $\tau_{DIV} \approx 13$ hr, measured previously for the same cell line (R1) and growth conditions (serum/LIF) (**Cartwright et al., 2005**). Using this calculation, the measured DNA content for each cell was converted to time within the cell cycle.

### 7.2 Estimating the gene replication time and the degree of dosage compensation

Determining the time within the cell cycle for each cell allowed us to determine whether there are changes in transcription activity over time. In particular, we wanted to refine the measurement of fold-change in nascent mRNA following gene replication (**Figure 2B**), and to estimate the gene replication time. To do so, we plotted the number of nascent mRNA $n$ against the calculated time within the cell cycle $t$ (**Figure 3B**). The individual values of nascent mRNA were smoothed by averaging over the nearest 50 cells in time. Using the fit routine in MATLAB, the smoothed data were fitted to a piecewise function of the form:

$$n(t) = \begin{cases} 2\beta, & t < \tau_{REP} \\ 4\eta\beta, & t \geq \tau_{REP} \end{cases}$$

where $\beta$ describes the average number of nascent mRNA produced per gene copy and $\tau_{REP}$ is the gene replication time. Dosage compensation is included using the parameter $\eta$, the fold-change in nascent mRNA per gene copy following gene replication.

## 8. A cell-cycle dependent stochastic model of gene activity

### 8.1 Description of the model

Our model is built on the two-state model commonly used in the literature (**Raj et al., 2006**; **So et al., 2011**; **Zong et al., 2010**), but is extended to explicitly include two additional features: nascent (actively transcribed) mRNA, and the cell-cycle effects of gene replication and dosage compensation. In this model (**Figure 3A**), each gene copy stochastically switches between the 'OFF' and 'ON' states with rates $k_{OFF}$ and $k_{ON}$, and transcription is stochastically initiated only in the 'ON' state with rate $k_{INI}$. After transcription has been initiated, the nascent transcript elongates and remains at the transcription site for a total residence time, $\tau_{RES}$. After time $\tau_{RES}$, the nascent mRNA is released and converted into a mature mRNA. Mature mRNA is then degraded stochastically with rate $k_D$. At the gene replication time in the cell cycle $\tau_{REP}$, the gene copy number doubles from two to four. The effect of dosage compensation—decreased transcription following gene replication—is included

through a fold-change $\alpha$ in $k_{ON}$, where $\alpha<1$ (invoking instead a change in $k_{OFF}$ following gene replication does not significantly change the fitting results; *Figure 3—figure supplement 5*). At the cell division time $\tau_{DIV}$, the mature mRNA are binomially partitioned to the two daughter cells.

## 8.2 Solving the model

We first note that the deterministic lifetime of nascent mRNA represents a constant 'time delay' before nascent mRNA is converted into mature mRNA. Given that this time delay is short compared to the duration of cell-cycle phases, the mature mRNA distribution can be approximated using a model where mature mRNA is produced immediately upon an initiation event. Below, the mature and nascent mRNA distributions were therefore calculated separately while sharing the same 'ON'/ 'OFF' switching and transcription initiation kinetics.

### 8.2.1 Calculating the mature mRNA distributions

To calculate the mature mRNA distributions in consideration of the gene replication process (from one copy to two copies within a cell cycle), our approach was to include two gene copies throughout the cell cycle, where the second gene copy remains in the 'OFF' state until the gene replication time. To start, we first defined the joint probability at time $t$ as:

$$P_{s_1,s_2}(m,t),$$

where the states for each of the two gene copies, $s_1$ and $s_2$, can be 'ON' (denoted as 1) or 'OFF' (denoted as 0), and the number of mRNA, $m$, is a nonnegative integer (0,1,2,...).

We then constructed the probability vector $\mathbf{P}(t)$, which contains the probabilities of all possible states at time $t$:

$$\mathbf{P}(t) = \begin{bmatrix} \mathbf{P}(0,t) \\ \mathbf{P}(1,t) \\ \vdots \\ \mathbf{P}(m,t) \\ \vdots \end{bmatrix}, \; where \; \mathbf{P}(m,t) = \begin{bmatrix} P_{0,0}(m,t) \\ P_{1,0}(m,t) \\ P_{0,1}(m,t) \\ P_{1,1}(m,t) \end{bmatrix}$$

The vector $\mathbf{P}(m,t)$ contains the probabilities of the states that have exactly $m$ mRNA at time $t$.

Next, we constructed the Chemical Master Equation (CME), the series of ordinary differential equations that describes the rate of change of these probabilities in time. The CME can be written as:

$$\frac{d}{dt}\mathbf{P}(t) = \mathbf{Q}(t)\mathbf{P}(t),$$

where $\mathbf{Q}(t)$ denotes the rates of transition between states. $\mathbf{Q}(t)$ is time-dependent, reflecting the differences in gene-state transitions from before- to after gene replication. In particular, the second gene copy is allowed to transition to the 'ON' state after the gene replication time. In our model, $\mathbf{Q}(t)$ is constructed as:

$$\mathbf{Q}(t) = \begin{bmatrix} \mathbf{A}(t)-\mathbf{T} & \mathbf{D} & 0 & \cdots \\ \mathbf{T} & \mathbf{A}(t)-\mathbf{T}-\mathbf{D} & 2\mathbf{D} & \ddots \\ 0 & \mathbf{T} & \mathbf{A}(t)-\mathbf{T}-2\mathbf{D} & \ddots \\ \vdots & \ddots & \ddots & \ddots \end{bmatrix}.$$

In this expression, $\mathbf{A}(t)$ is the gene-state transition matrix, $\mathbf{T}$ is the transcription matrix, and $\mathbf{D}$ is the degradation matrix, defined as follows:

$$\mathbf{A}(t) = \begin{cases} \begin{bmatrix} -k_{ON} & k_{OFF} & 0 & 0 \\ k_{ON} & -k_{OFF} & 0 & 0 \\ 0 & 0 & 0 & 0 \\ 0 & 0 & 0 & 0 \end{bmatrix} & t < \tau_{REP} \\[2em] \begin{bmatrix} -2\alpha k_{ON} & k_{OFF} & k_{OFF} & 0 \\ \alpha k_{ON} & -\alpha k_{ON} - k_{OFF} & 0 & k_{OFF} \\ \alpha k_{ON} & 0 & -\alpha k_{ON} - k_{OFF} & k_{OFF} \\ 0 & \alpha k_{ON} & \alpha k_{ON} & -2k_{OFF} \end{bmatrix} & t \geq \tau_{REP} \end{cases} ;$$

$$\mathbf{T} = \begin{bmatrix} 0 & 0 & 0 & 0 \\ 0 & k_{INI} & 0 & 0 \\ 0 & 0 & k_{INI} & 0 \\ 0 & 0 & 0 & 2k_{INI} \end{bmatrix} ; \quad \mathbf{D} = \begin{bmatrix} k_D & 0 & 0 & 0 \\ 0 & k_D & 0 & 0 \\ 0 & 0 & k_D & 0 \\ 0 & 0 & 0 & k_D \end{bmatrix},$$

where $\alpha$ is the fold-change of the gene activation rate $k_{ON}$ following gene replication. The value of $\alpha$ is calculated from the fold-change in nascent mRNA per gene copy $\eta$ using the relation (see Materials and methods 8.4 for the derivation):

$$\alpha = \frac{\eta k_{OFF}}{(1-\eta)k_{ON} + k_{OFF}}.$$

Note that $\mathbf{Q}(t)$ was constructed such that it changes at the gene replication time, but is constant at all other times.

The CME represents an infinite number of ordinary differential equations because $m$ can be any nonnegative integer. We followed the Finite State Projection (FSP) approach (**Munsky and Khammash, 2006**) and truncated the system to a finite number of $m$, enabling the numerical calculation of solutions to the model. The chance of observing >1300 mature mRNA in a cell is very low (<1 cell per 5000 cells), so we set the truncation value to $m$ = 1500.

To numerically calculate the model solution for a given set of parameters $\{k_{ON}, k_{OFF}, k_{INI}, k_D, \tau_{REP}, \alpha\}$, we implemented the following algorithm in MATLAB:

1) The vector $\mathbf{P}(t)$ was initialized at time $t = 0$. For simplicity, we initialized the system to have $m$ = 0, $s_1$ = 0, $s_2$ = 0:

$$\mathbf{P}(0) = \begin{bmatrix} \mathbf{P}(0,0) \\ \mathbf{P}(1,0) \\ \vdots \end{bmatrix} = \begin{bmatrix} \begin{bmatrix} P_{0,0}(0,0) \\ P_{1,0}(0,0) \\ P_{0,1}(0,0) \\ P_{1,1}(0,0) \end{bmatrix} \\ \begin{bmatrix} P_{0,0}(1,0) \\ P_{1,0}(1,0) \\ P_{0,1}(1,0) \\ P_{1,1}(1,0) \end{bmatrix} \\ \vdots \end{bmatrix} = \begin{bmatrix} \begin{bmatrix} 1 \\ 0 \\ 0 \\ 0 \end{bmatrix} \\ \begin{bmatrix} 0 \\ 0 \\ 0 \\ 0 \end{bmatrix} \\ \vdots \end{bmatrix}.$$

2) $\mathbf{P}(0)$ was then time-propagated to the gene replication time $\tau_{REP}$. For a $\mathbf{Q}$ that is constant in time, time-propagation of the CME from $t = \tau_1$ to $t = \tau_2$ can be calculated using the exponential operator: $\mathbf{P}(\tau_2) = \exp\{\mathbf{Q}[\tau_2 - \tau_1]\}\mathbf{P}(\tau_1)$. A direct implementation in MATLAB uses the expm function. However, we found that the large size of $\mathbf{Q}$ in our case resulted in a prohibitively slow calculation. To balance accuracy and speed, we instead approximated the previous calculation with a series of discreet time-propagation steps: $\mathbf{P}(\tau_2) \approx \mathbf{P}_{dis}(\tau_2) = (\mathbf{I} + \mathbf{Q}\Delta t)^{\frac{\tau_2 - \tau_1}{\Delta t}}\mathbf{P}(\tau_1)$, where $\mathbf{P}_{dis}(\tau_2)$ is the approximated result, $\mathbf{I}$ is the identity matrix, and $\Delta t$ is the time interval of each time-propagation step. We found that by setting $\Delta t$=0.001 min, the calculation could be performed in less than a second with little deviation from the result of the exponential operator for all sets of parameters used ($\sum_m |\mathbf{P}_{dis}(\tau_2) - \mathbf{P}(\tau_2)| < 10^{-6}$). We therefore used the series of discreet time-propagation steps when computing model solutions.

To time-propagate $\mathbf{P}(0)$ to the gene replication time $\tau_{REP}$, we used the following operation:

$$\mathbf{P}(\tau_{REP}^-) = (\mathbf{I} + \mathbf{Q}(t < \tau_{REP})\Delta t)^{\frac{\tau_{REP}}{\Delta t}}\mathbf{P}(0),$$

where $\mathbf{P}(\tau_{REP}^-)$ represents the probability vector at time $\tau_{REP}$, before the operation performed in 3).

3) At the gene replication time $\tau_{REP}$, the second gene copy was assigned the gene state of the first gene copy (recall that until $\tau_{REP}$, the second gene copy remained in the 'OFF' state: $P_{0,1}(m,t) = P_{1,1}(m,t) = 0$). To accomplish this, we constructed the gene replication operator $\mathbf{R}$, as follows:

$$\mathbf{R} = \begin{bmatrix} \mathbf{Rm} & 0 & 0 & \cdots \\ 0 & \mathbf{Rm} & 0 & \ddots \\ 0 & 0 & \mathbf{Rm} & \ddots \\ \vdots & \ddots & \ddots & \ddots \end{bmatrix}, \text{ where } \mathbf{Rm} = \begin{bmatrix} 1 & 0 & 0 & 0 \\ 0 & 0 & 0 & 0 \\ 0 & 0 & 0 & 0 \\ 0 & 1 & 0 & 0 \end{bmatrix}.$$

The operation $\mathbf{P}(\tau_{REP}^{+}) = \mathbf{RP}(\tau_{REP}^{-})$ was performed such that, for each $m$:

$$\mathbf{P}(m, \tau_{REP}^{+}) = \begin{bmatrix} P_{0,0}(m,\tau_{REP}^{+}) \\ P_{1,0}(m,\tau_{REP}^{+}) \\ P_{0,1}(m,\tau_{REP}^{+}) \\ P_{1,1}(m,\tau_{REP}^{+}) \end{bmatrix} = \begin{bmatrix} P_{0,0}(m,\tau_{REP}^{-}) \\ 0 \\ 0 \\ P_{1,0}(m,\tau_{REP}^{-}) \end{bmatrix},$$

where $\mathbf{P}(\tau_{REP}^{-})$ and $\mathbf{P}(\tau_{REP}^{+})$ represent the probability vectors before and after the application of $\mathbf{R}$ at time, $\tau_{REP}$, respectively.

4) $\mathbf{P}(\tau_{REP}^{+})$ was then time-propagated to the cell division time, $\tau_{DIV}$, using the operation:

$$\mathbf{P}(\tau_{DIV}^{-}) = (\mathbf{I} + \mathbf{Q}(t \geq \tau_{REP})\Delta t)^{\frac{\tau_{DIV} - \tau_{REP}}{\Delta t}} \mathbf{P}(\tau_{REP}^{+}),$$

where $\mathbf{P}(\tau_{DIV}^{-})$ represents the probability vector at time, $\tau_{DIV}$, before the operation performed in 5).

5) At the cell division time $\tau_{DIV}$, the mRNA were binomially partitioned, and the second gene copy was transitioned to the 'OFF' state. To perform these two operations, we constructed a binomial partitioning operator $\mathbf{B}$ and a cell division operator $\mathbf{V}$, defined as follows:

$$\mathbf{B} = \begin{bmatrix} B(0|0)\mathbf{I} & B(0|1)\mathbf{I} & B(0|2)\mathbf{I} & \cdots \\ 0 & B(1|1)\mathbf{I} & B(1|2)\mathbf{I} & \ddots \\ 0 & 0 & B(2|2)\mathbf{I} & \ddots \\ \vdots & \ddots & \ddots & \ddots \end{bmatrix}, \text{ where } B(m|k) = \binom{k}{m} 2^{-k}, \mathbf{I} = \begin{bmatrix} 1 & 0 & 0 & 0 \\ 0 & 1 & 0 & 0 \\ 0 & 0 & 1 & 0 \\ 0 & 0 & 0 & 1 \end{bmatrix};$$

$$\mathbf{V} = \begin{bmatrix} \mathbf{Vm} & 0 & 0 & \cdots \\ 0 & \mathbf{Vm} & 0 & \ddots \\ 0 & 0 & \mathbf{Vm} & \ddots \\ \vdots & \ddots & \ddots & \ddots \end{bmatrix}, \text{ where } \mathbf{Vm} = \begin{bmatrix} 1 & 0 & 1 & 0 \\ 0 & 1 & 0 & 1 \\ 0 & 0 & 0 & 0 \\ 0 & 0 & 0 & 0 \end{bmatrix}.$$

Note that $\mathbf{B}$ and $\mathbf{V}$ commute ($\mathbf{BV}=\mathbf{VB}$), so the order in which they are applied does not affect the result. The operation $\mathbf{P}(\tau_{DIV}^{+}) = \mathbf{BVP}(\tau_{DIV}^{-})$ was applied such that, for each $m$:

$$\mathbf{P}(m, \tau_{DIV}^{+}) = \begin{bmatrix} P_{0,0}(m,\tau_{DIV}^{+}) \\ P_{1,0}(m,\tau_{DIV}^{+}) \\ P_{0,1}(m,\tau_{DIV}^{+}) \\ P_{1,1}(m,\tau_{DIV}^{+}) \end{bmatrix} = \begin{bmatrix} \sum_{i,k} B(m|k)P_{0,i}(m,\tau_{DIV}^{-}) \\ \sum_{i,k} B(m|k)P_{1,i}(m,\tau_{DIV}^{-}) \\ 0 \\ 0 \end{bmatrix},$$

where $\mathbf{P}(\tau_{DIV}^{-})$ and $\mathbf{P}(\tau_{DIV}^{+})$ represent the probability vectors before and after the application of $\mathbf{B}$ and $\mathbf{V}$ at time $\tau_{DIV}$, respectively.

6) The resulting vector $\mathbf{P}(\tau_{DIV}^{+})$ was next compared to $\mathbf{P}(0)$ for indication of a cyclostationary solution (i.e. solutions that satisfy $\mathbf{P}(t) = \mathbf{P}(t + \tau_{DIV})$). If $\mathbf{P}(\tau_{DIV}^{+})$ did not satisfy the criterion $\sum_{m} \left| \mathbf{P}(\tau_{DIV}^{+}) - \mathbf{P}(0) \right| < 10^{-6}$, then $\mathbf{P}(\tau_{DIV}^{+})$ was assigned to $\mathbf{P}(0)$ and steps 2-6 were repeated. If $\mathbf{P}(\tau_{DIV}^{+})$ did satisfy the above criterion, it was used as $\mathbf{P}(0)$ of the solution to the model. The solution was then propagated through the above algorithm once again. During this final propagation, $\mathbf{P}(t)$ was recorded at 20 evenly spaced time points along the cell cycle.

### 8.2.2 Calculating the nascent mRNA distributions

To solve the model (*Figure 3A*) for the nascent mRNA distributions, we used the modified version of the FSP algorithm described in (*Xu et al., 2015*), which considers that nascent mRNA elongates at a constant rate and remains at the site of transcription for a deterministic residence time. This model explicitly considers the positions of smFISH probes along the gene. Here for simplicity we approximated these positions as distributed uniformly along the gene, because we label 4 (for *Nanog*) or 5 (for *Oct4*) exons, as well as the 3' UTR of the gene. We used this algorithm to calculate the distribution of nascent mRNA produced from a single gene copy at 20 evenly spaced time points in the cell cycle (identical to the evaluation times of the mature mRNA distributions). For times before gene replication, we used a given set of parameter values for the gene activation rate $k_{ON}$, the gene inactivation rate $k_{OFF}$, the transcription initiation rate $k_{INI}$ and the residence time $\tau_{RES}$. For times after gene replication, we modified the gene activation rate to $\alpha k_{ON}$, where $\alpha$ is calculated from the fold-change in nascent mRNA per gene copy $\eta$ using the relation (see Section 8.4 for derivation):

$$\alpha = \frac{\eta k_{OFF}}{(1-\eta)k_{ON} + k_{OFF}}.$$

### 8.2.3 Predicting the mRNA distributions corresponding to 2 and 4 gene copies

In the previous section, we solved for the mature and nascent mRNA distributions in the case where the cell cycle begins with a single gene copy present. To compare our model with the experimental data, we considered the actual gene copy number in the cell, namely two copies that replicate into four copies during the cell cycle. Assuming that the gene copies are independent of each other in terms of 'ON'/'OFF' switching and transcription initiation, which is supported by the experimental results for *Oct4* and *Nanog* (*Figure 2A*, *Figure 2—figure supplement 1*), the distribution of mRNA from multiple gene copies is equal to the autoconvolution of that from a single gene copy (*Bahar Halpern et al., 2015b*). Therefore, we solved for the mature mRNA distribution by calculating the autoconvolution of the model solution. We solved for the nascent mRNA distribution at times before gene replication by calculating the autoconvolution of the model solution, and solved for the nascent mRNA distribution at times after gene replication by performing two successive autoconvolution calculations of the model solution (the second calculation was performed on the output of the first).

The mature and nascent mRNA distributions obtained at this point were used to fit the experimental smFISH data using the procedures described in the following section.

### 8.3 Maximum likelihood estimation of model parameters

To determine the set of parameters that best fits the experimental data, we used the maximum likelihood estimation method, following (*Neuert et al., 2013*). Briefly, given data from $N$ cells, a likelihood function can be constructed which quantifies how likely it is that the data came from a given model and parameter set. To construct the likelihood function, we first calculated the probability, given the parameter set $\mathbf{K}$, of observing a cell with $m$ mature mRNA and $n$ nascent mRNA at time $t$:

$$P_{mat}(m,t|\mathbf{K})P_{nas}(n,t|\mathbf{K}),$$

where $P_{mat}$ and $P_{nas}$ are the probability distributions predicted by the model for mature and nascent mRNA, respectively. The likelihood function $L$ describes the total probability of observing the $N$ data points given the model parameter set $\mathbf{K}$:

$$L(\mathbf{K}) = \prod_{i=1}^{N} P_{mat}(m_i,t_i|\mathbf{K})P_{nas}(n_i,t_i|\mathbf{K}).$$

The parameter set that maximizes the likelihood function (which also maximizes the logarithm of the likelihood function) produces the best model fit to the experimental data:

$$\mathbf{K}_{Fit} = \arg\max_{\mathbf{K}}(\log(L(\mathbf{K}))) = \arg\max_{\mathbf{K}}\left(\sum_{i=1}^{N}\log(P_{mat}(m_i,t_i|\mathbf{K})P_{nas}(n_i,t_i|\mathbf{K}))\right).$$

In our model, $\mathbf{K}$ is comprised of fitting parameters $\{k_{ON}, k_{OFF}, k_{INI}, 1/\tau_{RES}\}$, parameters measured for each experiment $\{\tau_{REP}, \alpha\}$, and parameters from literature $\{k_D, \tau_{DIV}\}$. To find $\mathbf{K}_{Fit}$, we first computed libraries of $P_{mat}(m, t|\mathbf{K})$ and $P_{nas}(n, t|\mathbf{K})$ for each experiment. The libraries consist of model predictions for ranges of values for fit parameters $\{k_{ON}, k_{OFF}, k_{INI}, 1/\tau_{RES}\}$, where each parameter samples the biologically plausible rates ($10^{-3}$-$10^2$ min$^{-1}$ (*Sanchez et al., 2013*), with log-intervals of $10^{0.2}$ min$^{-1}$). $\tau_{REP}$ was measured for each experiment (Materials and methods 7.2). $\alpha$ was calculated based on the value of $\eta$ measured for each experiment (Materials and methods 7.2). $k_D$ was taken as the mean of the known literature values (*Abranches et al., 2013*; *Muñoz Descalzo et al., 2013*; *Ochiai et al., 2014*; *Sharova et al., 2009*) (*Supplementary file 1*). $\tau_{DIV}$ was taken from the literature (*Cartwright et al., 2005*).

To compare each data point to the model, the number of nascent and mature mRNA was rounded up or down to the nearest integer. The time in the cell cycle was rounded up or down to the nearest of the 20 time points at which the model was solved. Then, for each experiment and corresponding library, the likelihood value was evaluated for all parameter sets. The maximum likelihood value was determined and used as an estimate of the optimal parameter set. We then refined each fit library to scan $10^{-0.5}$–$10^{0.5}$ min$^{-1}$ fold of the previous estimate at a finer resolution of $10^{0.025}$ min$^{-1}$, and searched for the maximum likelihood value. The parameters that produced the maximum likelihood value were taken to be $\mathbf{K}_{Fit}$, and are shown in *Figure 3D*.

## 8.4 Converting fold-change in nascent mRNA to fold-change in $k_{ON}$ following gene replication

To include dosage compensation through a decrease in $k_{ON}$, we needed to find a mapping between the measured fold-change in number of nascent mRNA per gene copy following gene replication $\eta$ to the fold-change in $k_{ON}$ following gene replication $\alpha$. We started with the expression for mean number of nascent mRNA in the cell $\langle n \rangle$, which follows from (*Xu et al., 2015*):

$$\langle n(t) \rangle = \lambda \frac{g(t)k'_{ON}(t)k_{INI}\tau_{RES}}{(k'_{ON}(t) + k_{OFF})}.$$

This expression can be understood as the product of the following terms: (1) The fraction of time the gene is 'ON' ($k_{ON}/(k_{ON} + k_{OFF})$). (2) The rate of initiation when the gene is 'ON' ($k_{INI}$). (3) The time a nascent mRNA molecule spends on the gene ($\tau_{RES}$). (4) The number of genes in the cell ($g$). (5) The effective number each nascent transcript contributes to the average ($\lambda$; reflecting that nascent transcripts can be observed partially elongated [*Senecal et al., 2014*; *Xu et al., 2015*]).

At the gene replication time in the cell cycle $\tau_{REP}$, the gene copy number doubles from 2 to 4:

$$g(t) = \begin{cases} 2 & t < \tau_{REP} \\ 4 & t \geq \tau_{REP} \end{cases}.$$

In our model, the effect of dosage compensation—decreased transcription frequency following gene replication—is included through a fold-change $\alpha$ in $k_{ON}$, where $\alpha < 1$:

$$k'_{ON}(t) = \begin{cases} k_{ON} & t < \tau_{REP} \\ \alpha k_{ON} & t \geq \tau_{REP} \end{cases}.$$

To compare to the measured fold-change of nascent mRNA following gene replication (Materials and methods 7.2), we solved for the ratio of the expected means of nascent mRNA:

$$\frac{\langle n(t \geq \tau_{REP}) \rangle}{\langle n(t < \tau_{REP}) \rangle} = 2\alpha \frac{k_{ON} + k_{OFF}}{\alpha k_{ON} + k_{OFF}} = 2\eta$$

From this expression, we can obtain the mapping from the measured fold-change in nascent mRNA per gene copy following gene replication $\eta$ to the fold-change in rate of gene activation following gene replication $\alpha$:

$$\alpha = \frac{\eta k_{OFF}}{(1 - \eta)k_{ON} + k_{OFF}}.$$

## 9. A deterministic model for nascent and mature mRNA kinetics

To examine how the observed ratios of both nascent and mature mRNA numbers before/after gene replication are affected by the relative timescales of mRNA lifetime and cell cycle duration, we created a simple deterministic model for the kinetics of the two species. The model includes only mRNA production and degradation, along with the cell-cycle effects of gene replication and cell division, but disregarding gene-state switching and dosage compensation. The level of each mRNA species is described by:

$$\frac{d}{dt}R(t) = g(t)k_{INI} - k_D R(t), \; g(t) = \begin{cases} 1 & 0 \leq t < \tau_{REP} \\ 2 & \tau_{REP} \leq t < \tau_{DIV} \end{cases},$$

where $R(t)$ and $g(t)$ are the mRNA and gene copy-numbers, $k_{INI}$ and $k_D$ are the rates of mRNA transcription and degradation, $\tau_{REP}$ and $\tau_{DIV}$ are the times of gene replication and cell division. When solving for nascent mRNA using this formalism, an effective degradation rate is used, which corresponds to the residence time at the gene, $k_D = 1/\tau_{RES}$. At the end of the cell cycle, mRNA are partitioned to the daughter cells. To obtain the cyclostationary solution, we imposed the boundary condition $R(\tau_{DIV}) = 2R(0)$. The solution is the following piecewise function:

$$R(t) = \begin{cases} \dfrac{k_{INI}}{k_D}\left[1 - \dfrac{e^{-k_D(\tau_{DIV}-\tau_{REP})}}{2 - e^{-k_D \tau_{DIV}}}e^{-k_D t}\right] & 0 \leq t < \tau_{REP} \\[3ex] \dfrac{k_{INI}}{k_D}\left[2 - e^{-k_D(t-\tau_{REP})} - \dfrac{e^{-k_D(\tau_{DIV}-\tau_{REP})}}{2 - e^{-k_D \tau_{DIV}}}e^{-k_D t}\right] & \tau_{REP} \leq t < \tau_{DIV} \end{cases}.$$

$R(t)$ is plotted in *Figure 3—figure supplement 1A* for both mature and nascent *Oct4* mRNA using the measured gene replication time ($\tau_{REP}$; *Figure 3—figure supplement 4*), the effective transcription initiation rate from averaging over 'ON'/'OFF' gene states ($k_{INI}$=0.6 min⁻¹; *Figure 3D*), the literature average of mature mRNA degradation rate ($k_D$; *Supplementary file 1*), the measured residence time ($\tau_{RES}$; *Figure 3D*), and the literature estimate of the cell division time ($\tau_{DIV}$=13 hr; *[Cartwright et al., 2005]*).

Next, we defined observation time windows for the early and late parts of the cell cycle, within which the numbers of mRNA are averaged:

$$\langle R(0 \leq t < \tau_1) \rangle = \frac{1}{\tau_1}\int_0^{\tau_1} dt\, R(t),$$

$$\text{and } \langle R(\tau_2 \leq t < \tau_{DIV}) \rangle = \frac{1}{\tau_{DIV} - \tau_2}\int_{\tau_2}^{\tau_{DIV}} dt\, R(t),$$

where $\tau_1$ is some time in the beginning of the cell cycle before the gene replication time, and $\tau_2$ is some time near the end of the cell cycle after the gene replication time. The ratio, $R_M$, is defined as:

$$R_M \equiv \frac{\langle R(\tau_2 \leq t < \tau_{DIV}) \rangle}{\langle R(0 \leq t < \tau_1) \rangle}.$$

We calculated $R_M$ for nascent and mature *Oct4* mRNA (*Figure 3—figure supplement 1B*) using the periods of G1 and G2 phases extracted from the cell cycle model (*Figure 1F*) as the first ($0 \leq t < \tau_1$) and second ($\tau_2 \leq t < \tau_{DIV}$) observation time windows in addition to the parameters used above. To demonstrate the effect of varying mRNA lifetimes, we plotted $R_M$ against $k_D \tau_{DIV}$ (*Figure 3—figure supplement 1C*).

## Acknowledgements

We are grateful to the following people for generous advice and for providing reagents: M Dejosez, L McLane, A Raj, L Sepulveda, L-H So, A Sokac, M Wang and A Warmflash. Work in the Golding lab is supported by grants from NIH (R01 GM082837), NSF (PHY 1147498, PHY 1430124 and PHY 1427654), The Welch Foundation (Q-1759) and The John S. Dunn Foundation (Collaborative

Research Award). H Xu holds a Burroughs Wellcome Fund Career Award at the Scientific Interface. Work in the Zwaka lab is supported by the Huffington Foundation and by grants from the NIH (R01 GM077442 and P01 GM81627). We gratefully acknowledge the computing resources provided by the CIBR Center of Baylor College of Medicine.

## Additional information

### Funding

| Funder | Grant reference number | Author |
|---|---|---|
| John S. Dunn Foundation | Collaborative Research Award | Samuel O Skinner<br>Heng Xu<br>Ido Golding |
| Burroughs Wellcome Fund | Career Award at the Scientific Interface | Heng Xu |
| Huffington Foundation | | Thomas P Zwaka |
| National Institutes of Health | R01 GM077442 | Thomas P Zwaka |
| National Institutes of Health | P01 GM81627 | Thomas P Zwaka |
| National Science Foundation | PHY 1427654 | Ido Golding |
| National Institutes of Health | R01 GM082837 | Ido Golding |
| National Science Foundation | PHY 1147498 | Ido Golding |
| Welch Foundation | Q-1759 | Ido Golding |
| National Science Foundation | PHY 1430124 | Ido Golding |

The funders had no role in study design, data collection and interpretation, or the decision to submit the work for publication.

### Author contributions

SOS, Conceived the experimental and analysis methods, Developed image and data analysis algorithms and theoretical models, Developed the smFISH protocol, Performed the cell culture, Performed the experiments, Developed algorithms and theoretical models, Acquisition of data, Analysis and interpretation of data, Drafting or revising the article; HX, Developed image and data analysis algorithms and theoretical models, Analysis and interpretation of data, Contributed unpublished essential data or reagents; SNJ, Developed the smFISH protocol, Performed the cell culture, Acquisition of data, Contributed unpublished essential data or reagents; PRF, Performed the cell culture, Acquisition of data, Contributed unpublished essential data or reagents; TPZ, Conceived the experimental and analysis methods, Provided guidance on stem cell biology, Supervised the project, Conception and design, Drafting or revising the article, Contributed unpublished essential data or reagents; IG, Conceived the experimental and analysis methods, Supervised the project, Conception and design, Analysis and interpretation of data, Drafting or revising the article

### Author ORCIDs

Ido Golding, http://orcid.org/0000-0002-4308-4959

## Additional files

### Supplementary files

• Supplementary file 1. Literature estimates of transcription parameters used in this study.

• Supplementary file 2. Estimated parameters of transcription for *Oct4* and *Nanog*.

• Supplementary file 3. Sequences of smFISH probes.

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
