## [Decision Letter]

Thank you for submitting your work entitled "Single-cell analysis of transcription
kinetics across the cell cycle" for consideration by *eLife*. Your
article has been reviewed by two peer reviewers, and the evaluation has been overseen by
Rob Singer (Reviewing Editor) and Aviv Regev as the Senior Editor.

The reviewers have discussed the reviews with one another and the Reviewing Editor has
drafted this decision to help you prepare a revised submission.

As you can see below, the reviewers were enthusiastic about the manuscript and felt it
was of high quality. The need for revision centers on some items where they felt the
presentation required more extensive discussion, for instance in the dosage compensation
discussion and the modeling approach. We look forward to the revised manuscript
soon.

*Reviewer #1:*

In the lovely paper "Single-cell analysis of transcription kinetics across the cell
cycle" by Skinner et al., the authors investigate how transcriptional parameters of
Nanog and Oct4 affect the cell-to-cell variability of these genes and how these
parameters change during the cell cycle. Using single-molecule FISH measurement to
precisely quantify nascent and mature RNA, and by determining the transcriptional
kinetic parameters the authors show that the difference in variability between the two
genes can be explained by the slower ON/OFF switching by Nanog.

I think this study is very timely, as there has been increased interest these days in
the connections between global regulation of transcriptional processes and
transcriptional bursts-in this case, the demonstration that there is dosage compensation
upon DNA replication. The authors also have wisely chosen to study Nanog and Oct4, which
has been the topic of much recent debate. One of the highlights is the authors showing
that the kinetics of Nanog are what leads to the oft-described variability in Nanog
transcript levels. It is also methodologically rigorous, including the RNA
quantification, the modelling of the kinetic parameters the analysis, as well as the
extensive documentation of the methods used.

1) The more familiar usage of the term dosage compensation comes from the case of sex
chromosome dosage compensation (e.g., to balance out X chromosome dosage differences
between male and female mice). I think what the authors are observing is rightly called
dosage compensation, but it's probably worth mentioning the more traditional context in
which the term is used and explicitly pointing out the similarities and differences.

2) The paper was exceptional in its depth of methods documentation, yet regarding the
cell cycle modelling and the transcriptional kinetic parameters, the paper would benefit
if the authors described some of the modeling more in the main text. For example, it
would be useful to better clarify the difference between the "rough" and
detailed cell-cycle analysis, possibly in a sentence at the beginning of the section.
Similarly, it would be helpful if a brief explanation of the ergodic rate analysis could
also be found in the main text. Along these lines: Would be helpful to define the term
"cell cycle age". Also, in Figure 3,
there is no indication as to what the start and end point for "Time in cell
cycle" is, and thus how the 10 time windows relate to G1, S, G2 phase.

3) One of the results I found most interesting was that the reporter did not show any
dosage compensation effect. I was hoping the authors could speculate on this a bit more.
In the case of Padovan-Merhar et al., they show that whatever the cause is for the
dosage compensation, it's occurring in cis to the DNA, like a histone modification or
something that gets diluted upon replication. It's possible that the reporter gene is
not fully chromatinized, which is why it doesn't show the dosage compensation effect.
Anyway, I thought it was a cool result that the authors may want to highlight more.

Reviewer #2:

Cell cycle phase is one of the most important extrinsic factors determining differences
within populations of actively dividing cells. In this study Golding and colleagues
combine high-quality single molecule FISH of mature and nascent mRNA and computational
approaches to infer cell cycle phase and study its effect on changes in promoter burst
parameters. They demonstrate their approach by identifying a dosage compensation
mechanism entailing a decline in the burst frequency of the genes Nanog and Oct4. The
power of this work is in the rigorous and elegant theoretical formulation of the problem
of inferring burst parameters in cycling cells, and the clear description of the
algorithm for extracting these parameters. I believe the methodology developed here will
be instrumental to many future works related to gene expression variability in the
context of the cell cycle.

The paper could be improved by addressing, at least in the text, the following
points:

1) The authors should elaborate on the comparison between their results and those of Raj
and colleagues (Padovan-Merhar et al., 2015). Specifically in the Padovan-Merhar paper a
dosage compensation very similar to the one identified here was detected (decreased
"burst frequency" upon replication), however, upon growth of cellular volume
(occurring predominantly at G2) there was a global increase in number of nascent mRNA
per transcription site (compensatory increase in "burst size"). The present
study did not identify a difference in the burst size between G1 and G2. These
discrepancies between the two works could be related to the differences in the cell
lines and genes studied (specifically the shorter cell cycle time of ES cells compared
to fibroblasts).

2) The deterministic model of nascent and mature mRNA kinetics (section 9) and the
associated Figure 3—figure supplement 1 nicely
demonstrate that the mature mRNA is not at steady state. More importantly, it shows that
the mature mRNA in G2 is less than twice the levels in G1(as also shown in Figure 3). This would mean that upon division the
levels of mature mRNA at the start of G1 phase of the next round would be smaller than
in the current round, and that mRNA will exponentially decline to zero with additional
cycles. This naturally cannot be the case and there must be some compensatory dosage
compensation somewhere along the cell cycle. While identifying this additional dosage
compensation mechanism is beyond the scope of the current work it is important to note
this issue in the text.

3) Section 6.1 “Quantification of DNA content”: the authors should provide the cell
cycle periods for the ES cells studied, inferred by their cell cycle phase inference
algorithm.

4) The authors consider a change in Kon upon replication, rather than Koff. One could
imagine the dosage compensation would entail higher Koff rather than lower Kon. Would
there be a potential identifiability problem in discerning between models that allow
changes in both Kon and Koff?

5) The model applied assumes fixed times of replication and division, how would results
change if these parameters were allowed to vary (that is if they were sampled from some
normal distribution)?

6) "The number of nascent mRNA at each active transcription site was quantified in
the exon-channel by dividing the integrated intensity by the integrated intensity of a
single-mRNA molecule (Materials and methods 5.1)". This approach may introduce some
bias that depends on the probe library design. If all probes target the first part of
the gene then any RNA polymerase will have a nascent mRNA attached to it that includes
the full complement of probes and thus has intensity equal to a full mature mRNA. If,
however, probes are equally spread along the gene, the average RNA polymerase will have
an mRNA with half of the library probes yielding a 'dimmer' dot. Correction for this
effect is described in Bahar Halpern et al. 2015 and is worth considering.

---

## [Author Response]

Reviewer 1:

*1) The more familiar usage of the term dosage compensation comes from the case
of sex chromosome dosage compensation (e.g., to balance out X chromosome dosage
differences between male and female mice). I think what the authors are observing is
rightly called dosage compensation, but it's probably worth mentioning the more
traditional context in which the term is used and explicitly pointing out the
similarities and differences.*

In the revised manuscript, we now refer to the traditional context of “dosage
compensation” upon first using the term. We have also expanded the discussion of our
findings on this matter vis-à-vis those of Padovan-Merhar et al. (2015).

*2) The paper was exceptional in its depth of methods documentation, yet
regarding the cell cycle modelling and the transcriptional kinetic parameters, the
paper would benefit if the authors described some of the modeling more in the main
text. For example, it would be useful to better clarify the difference between the
"rough" and detailed cell-cycle analysis, possibly in a sentence at the
beginning of the section. Similarly, it would be helpful if a brief explanation of
the ergodic rate analysis could also be found in the main text. Along these lines:
Would be helpful to define the term "cell cycle age". Also, in Figure 3, there is no indication as to what the
start and end point for "Time in cell cycle" is, and thus how the 10 time
windows relate to G1, S, G2 phase.*

We have revised the text in a number of places to address the points highlighted by the
reviewer. Specifically: (i) We clarify the distinction between the “rough” cell-cycle
analysis, which consists of classifying cells into G1/S/G2, and the detailed one, which
calculates a specific time within the cell cycle and the Oct4/Nanog gene copy-number for
each cell. (ii) We define the function of the ergodic rate analysis. (iii) We omit the
ambiguous term “cell cycle age” previously used. (iv) We clarify how the 10 time windows
in Figure 3 are defined (caption to Figure 3).

*3) One of the results I found most interesting was that the reporter did not
show any dosage compensation effect. I was hoping the authors could speculate on this
a bit more. In the case of Padovan-Merhar et al., they show that whatever the cause
is for the dosage compensation, it's occurring in cis to the DNA, like a histone
modification or something that gets diluted upon replication. It's possible that the
reporter gene is not fully chromatinized, which is why it doesn't show the dosage
compensation effect. Anyway, I thought it was a cool result that the authors may want
to highlight more.*

We originally thought of this result merely as a control experiment, serving to validate
the observation of dosage compensation for *Oct4* and
*Nanog*. Following the reviewer’s comment, we now briefly discuss the
result and speculate that the viral enhancer elements included in the CAG promoter (Niwa
et al., 1991) are more resistant to the regulatory mechanisms that create the
compensatory effect in endogenous genes.

Reviewer #2:

*1) The authors should elaborate on the comparison between their results and
those of Raj and colleagues (Padovan-Merhar et al., 2015). Specifically in the
Padovan-Merhar paper a dosage compensation very similar to the one identified here
was detected (decreased "burst frequency" upon replication), however, upon
growth of cellular volume (occurring predominantly at G2) there was a global increase
in number of nascent mRNA per transcription site (compensatory increase in
"burst size"). The present study did not identify a difference in the burst
size between G1 and G2. These discrepancies between the two works could be related to
the differences in the cell lines and genes studied (specifically the shorter cell
cycle time of ES cells compared to fibroblasts).*

As the reviewer noted, our finding that the dosage compensation following Oct4 and Nanog
gene replication was achieved through a decrease in the probability of each gene copy to
be active (approximately equivalent to a change in burst frequency) was very similar to
what was reported by Padovan-Merhar et al. (2015) (their Figure 6A). But as also noted
by the reviewer, these authors also found that the cell volume (independently of the
cell cycle phase) strongly affects the number of nascent mRNAs at each transcription
site (approximately equivalent to a change in burst size). We were unable to test for
such an effect in our system, because the degree of cell-to-cell variability in volume
within each cell-cycle phase was significantly smaller compared to Padovan-Merhar et al.
(2015): CV≈0.2 (our data, not shown) versus CV≈0.5 (their Figure S3B). We therefore
cannot comment on the reviewer’s hypothesis, that burst-size modulation is absent in our
system due to differences in cell type of gene identity. This point is now briefly
discussed in paragraph eleven, “Results & Discussion”.

*2) The deterministic model of nascent and mature mRNA kinetics (section 9) and
the associated Figure 3—figure supplement 1
nicely demonstrate that the mature mRNA is not at steady state. More importantly, it
shows that the mature mRNA in G2 is less than twice the levels in G1(as also shown in
Figure 3). This would mean that upon
division the levels of mature mRNA at the start of G1 phase of the next round would
be smaller than in the current round, and that mRNA will exponentially decline to
zero with additional cycles. This naturally cannot be the case and there must be some
compensatory dosage compensation somewhere along the cell cycle. While identifying
this additional dosage compensation mechanism is beyond the scope of the current work
it is important to note this issue in the text.*

As the reviewer noted, the level of mature mRNA does not reach steady state during the
cell cycle, because the lifetime of mature mRNA is comparable to the duration of
individual cell cycle phases. However, we should clarify that the solution displayed in
Figure 3—figure supplement 1 was obtained by
requiring cyclostationarity: The number of mRNA at the end of the cell cycle is twice
that at the beginning of the cycle. Therefore, we do not expect our model to exhibit the
compensatory dynamics described by the reviewer. The cyclostationary nature of our
solution is now clarified in the caption to Figure 3—figure supplement 1.

*3) Section 6.1 “Quantification of DNA content”: the authors should provide the
cell cycle periods for the ES cells studied, inferred by their cell cycle phase
inference algorithm.*

We regret this omission. These values are now provided in paragraph two, subheading
“Fitting the DNA-content distribution to a cell-cycle model and determining cell-cycle
phases”.

*4) The authors consider a change in Kon upon replication, rather than Koff. One
could imagine the dosage compensation would entail higher Koff rather than lower Kon.
Would there be a potential identifiability problem in discerning between models that
allow changes in both Kon and Koff?*

The reviewer is correct. The observed dosage-compensation effect is also consistent with
a change in *k*_OFF_. We originally chose to invoke a change in
*k*_ON_ as this seemed to us the most parsimonious way of
explaining the observed change in number of active sites (without an accompanying change
in the number of nascent mRNA per site). This choice was also motivated by the findings
in previous studies (Xu et al., 2015, Senecal et al., 2014), that
*k*_ON_ is modulated to vary expression level. But as the
reviewer pointed out, our data does not allow us to distinguish whether
*k*_ON_ or *k*_OFF_ (or a combination
of the two) changes following gene replication. Importantly, however, we found that our
estimation of the transcription parameters for Oct4 and Nanog is insensitive to the
choice between *k*_ON_ and *k*_OFF_ as
the mediators of dosage compensation. Refitting our data using a model with a
*k*_OFF_-only change following gene replication gives very
close results to the *k*_ON_-only model in terms of the
estimated kinetics, besides the expected change in the values of
*k*_ON_ and *k*_OFF_ themselves (new
Figure 3—figure supplement 5). We have now
revised the text on in subheading “Description of the model” and the legends of Figure 3—figure supplement 5 to reflect these
points.

*5) The model applied assumes fixed times of replication and division, how would
results change if these parameters were allowed to vary (that is if they were sampled
from some normal distribution)?*

The assumption of fixed times of gene replication and cell division was made for
simplicity of modeling. Naturally, these parameters are likely to vary across cells in
the population. We have performed some numerical interrogation of how such variability
would propagate into the observed mRNA statistics, but we feel that these preliminary
studies do not reach the quality required for inclusion in the manuscript. However, the
reviewer’s comment brings up a larger point: The analysis we present here is unlikely to
account for all the contributions to cell-cell variability in mRNA numbers. First, as
the reviewer pointed out, the parameters of the cell cycle may themselves be variable,
possibly contributing to mRNA heterogeneity. Second, beyond gene dosage and the cell
cycle, the stochastic kinetics of multiple processes along the life history of mRNA—
elongation, splicing, nuclear export, degradation and partition—likely contribute to
cell- to-cell variability in the numbers of both nascent and mature mRNA. Additional
work, both experimental and theoretical, is required to delineate the contribution of
these processes to mRNA heterogeneity. In the revised manuscript, we have added a
Discussion paragraph to highlight this point (paragraph thirteen, Results &
Discussion).

*6) "The number of nascent mRNA at each active transcription site was
quantified in the exon-channel by dividing the integrated intensity by the integrated
intensity of a single-mRNA molecule (Materials and methods 5.1)". This approach
may introduce some bias that depends on the probe library design. If all probes
target the first part of the gene then any RNA polymerase will have a nascent mRNA
attached to it that includes the full complement of probes and thus has intensity
equal to a full mature mRNA. If, however, probes are equally spread along the gene,
the average RNA polymerase will have an mRNA with half of the library probes yielding
a 'dimmer' dot. Correction for this effect is described in Bahar Halpern et al. 2015
and is worth considering.*

The model we used for calculating the probability distribution for the amount of nascent
mRNA is based on the one we introduced in an earlier publication (Xu et al., 2015). The
model explicitly takes into account the positions of smFISH probes along the gene, thus
addressing the point raised by the reviewer. In the case of Oct4 and Nanog, each probe
set covers 4 (for Nanog) or 5 (for Oct4) exons, as well as the 3’ UTR of the gene. We
considered this coverage close enough to uniform and therefore, for simplicity, we
modeled probe coverage of the gene as uniform in both cases. This point is now clarified
in subheading “Calculating the nascent mRNA distributions”.